# A Bayesian model for identifying hierarchically organised states in neural population activity

**Patrick Putzky**[1,2,3], **Florian Franzen**[1,2,3], **Giacomo Bassetto**[1,3], **Jakob H. Macke**[1,3]

[1]Max Planck Institute for Biological Cybernetics, Tübingen
[2]Graduate Training Centre of Neuroscience, University of Tübingen
[3]Bernstein Center for Computational Neuroscience, Tübingen
`patrick.putzky@gmail.com, florian.franzen@tuebingen.mpg.de`
`giacomo.bassetto@tuebingen.mpg.de, jakob@tuebingen.mpg.de`

## Abstract

Neural population activity in cortical circuits is not solely driven by external inputs, but is also modulated by endogenous states which vary on multiple time-scales. To understand information processing in cortical circuits, we need to understand the statistical structure of internal states and their interaction with sensory inputs. Here, we present a statistical model for extracting hierarchically organised neural population states from multi-channel recordings of neural spiking activity. Population states are modelled using a hidden Markov decision tree with state-dependent tuning parameters and a generalised linear observation model. We present a variational Bayesian inference algorithm for estimating the posterior distribution over parameters from neural population recordings. On simulated data, we show that we can identify the underlying sequence of population states and reconstruct the ground truth parameters. Using population recordings from visual cortex, we find that a model with two levels of population states outperforms both a one-state and a two-state generalised linear model. Finally, we find that modelling of state-dependence also improves the accuracy with which sensory stimuli can be decoded from the population response.

## 1 Introduction

It has long been recognised that the firing properties of cortical neurons are not constant over time, but that neural systems can exhibit multiple distinct firing regimes. For example, cortical circuits can be in a 'synchronised' state during slow-wave sleep, exhibiting synchronised fluctuations of neural excitability [1] or in a 'desynchronised' state in which firing is irregular. Neural activity in anaesthetised animals exhibits distinct states which lead to widespread modulations of neural firing rates and contribute to cross-neural correlations [2]. Changes in network state can be brought about through the influence of inter-area interactions [3] and affect communication between cortical and subcortical structures [4].

Given the strong impact of cortical states on neural firing [3, 5, 4], an understanding of the interplay between internal states and external stimuli is essential for understanding how populations of cortical neurons collectively process information. Multi-cell recording techniques allow to record neural activity from dozens or even hundreds of neurons simultaneously, making it possible to identify the signatures of underlying states by fitting appropriate statistical models to neural population activity.

It is thought that the state-dependence of neocortical circuits is not well described using a global bi-modal state. Instead, the structure of cortical states is more accurately described

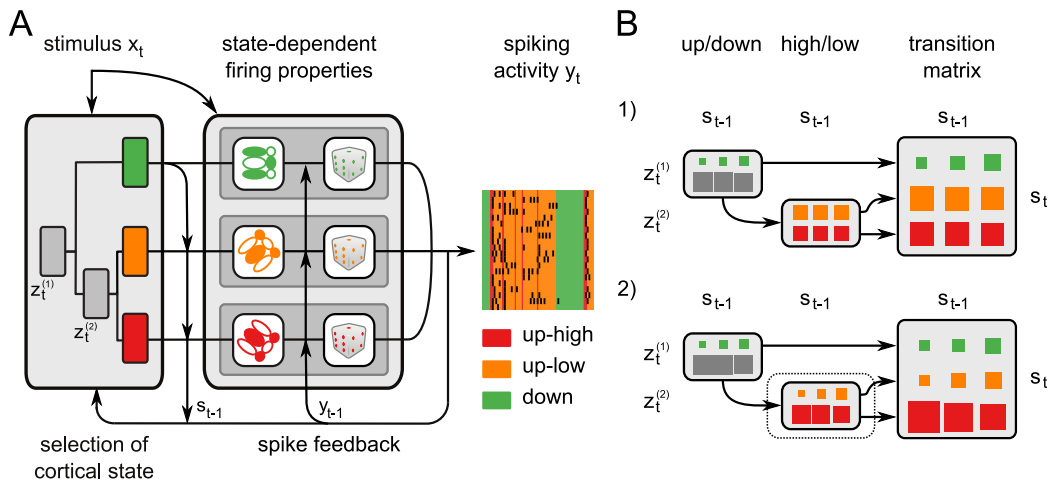

Figure 1: **Illustration of the model. A**) Generative model. At time $t$, the cortical state $s_t$ is determined using a Hidden Markov Decision Tree (HMDT) and depends on the previous state $\mathbf{s}_{t-1}$, population activity $\mathbf{y}_{t-1}$ and on the current stimulus $\mathbf{x}_t$. In our simulations, we assumed that the first split of the tree determined whether to transition into an up or down-state. Up-states contained transient periods of high firing across the population (up-high) as well as sustained periods of irregular firing (up-low). Each cortical state is then associated with different spike-generation dynamics, modelling state-dependence of firing properties such as 'burstiness'. **B)** State-transition probabilities depend on the tree-structure. Transition matrices are depicted as Hinton diagrams where each block represents a probability and each column sums to 1. Each row corresponds to the possible future state $s_t$ (see colour), and each column to the current state.
(1) A model in which transition-probabilities in the first level of the tree (up/down) are biased towards the up-state (green squares are bigger than gray ones), and weakly depend on the previous state $s_{t-1}$. In this example, both high/low phases are equally likely within up-states (second level of tree, depicted in second column) and do not depend on the previous state (all orange/red squares have same size). The resulting $3 \times 3$ matrix of transition probabilities across all states can be calculated from the transition-probabilities in the tree.
(2) Changing the properties of the second-level node only leads to a local change in the transition matrix: It affects the proportion between the orange/red states, but leaves the green state unchanged.

using multiple states which vary both between and within brain regions [6]. In addition, the 'state' of a neural population can vary across multiple time scales from milliseconds to seconds or more [6]: For example, cortical recordings can switch between up- and down-phases. During an up-phase cortical activity can exhibit 'volleys' of synchronised activity [7]—sometimes referred to as population bursts—which can be modelled as transient states.

These observations suggest that the structure of cortical states could be captured by a hierarchical organisation in which each state can give rise to multiple temporally nested 'sub-states'. This structure naturally yields a binary tree: States can be divided into sub-classes, with states further down the tree operating at faster time-scales determined by their parent node. We hypothesise that other cortical states also exhibit similar hierarchical structure. Our goal here is to provide a statistical model which can identify cortical states and their hierarchical organisation from recordings of population activity. As a running example of such a hierarchical organisation we use a model in which the population exhibits synchronised population bursts during up-states, but not during down-states. This system is modelled using a first level of state (up/down), and for which the up-state is further divided into two states (transient high-firing events and normal firing, see 1A).

We present an inhomogeneous hidden Markov model (HMM) [8] to model the temporal dynamics of state-transitions [9, 10]. Our approach is most closely related to [10], who developed a state-dependent generalised linear model [11] in which both the tuning prop-

erties and state-transitions can be modelled to depend on external covariates. However, our formulation also allows for hierarchically organised state-structures. In addition, previous population models based on discrete latent states [10, 12] used point-estimation for parameter learning. In contrast, we present algorithms for full Bayesian inference over the parameters of our model, making it possible to identify states in smaller or noisier data [13]. This is important for neural population recordings which are typically characterised by short recording times relative to the dimensionality of the data and by high variability. In addition, estimates of posterior distributions are important for visualising uncertainty and for optimising experimental paradigms with active-learning methods [14, 15].

## 2    Methods

We use a hidden Markov decision tree (HMDT) [16] to model hierarchically organised states with binary splits and a generalised linear observation model (GLM). An HMDT combines the properties of a hidden Markov model (to model temporal structure) with a hierarchical mixture of experts (HME, to model a hierarchy of latent states) [17]. In general the hierarchical approach can represent richer dependence of states on external covariates, analogous to the difference between multi-class logistic regression and multi-class binary decision trees. For example, a two-level binary tree  can separate four point clouds situated at the corners of a square whereas a 4-class multinomial regression cannot. We use Bayesian logistic regression [18] to model transition gates and emissions. In the following, we describe the model structure and propose a variational algorithm [8, 19] for inferring its parameters.

### 2.1    Hierarchical hidden Markov model for multivariate binary data

We consider discrete time-series data of multivariate binary[1] neural spiking events $\mathbf{y}_t \in \{0, 1\}^C$ where C is the number of cells. We assume that neural spiking can be influenced by (observed) covariates $\mathbf{x}_t \in \mathbb{R}^D$. The covariates $\mathbf{x}_t$ could represent external stimuli, spiking history of neurons or other measures such as the total population spike count. In our analyses below, we assume that correlations across neurons arise only from the joint coupling to the population state, and we do not include couplings between neurons as is sometimes done with GLMs [11]. Dependence of neural firing on internal states is modelled by including a 1-of-K latent state vector $\mathbf{s}_t$, where $K$ is the number of latent states. The emission probabilities for the observable vector $\mathbf{y}_t$ (i.e. the probability of spiking for each neuron) are thus given by

$$p\left(\mathbf{y}_t | \mathbf{x}_t, \mathbf{s}_t, \mathbf{\Phi}\right) = \prod_{i=1}^{K} \prod_{c=1}^{C} p\left(y_t^{(c)} | \mathbf{x}_t^{(c)}, \boldsymbol{\phi}_i^{(c)}\right)^{s_t^{(i)}}, \tag{1}$$

where $\mathbf{\Phi}$ is a set of model parameters. We allow the external covariate $\mathbf{x}_t$ to be different for each neuron $c$.

To model temporal dynamics over $\mathbf{s}_t$, we use a hidden Markov model (HMM) [10], where the state transitions take the form

$$p\left(\mathbf{s}_t | \mathbf{s}_{t-1}, \mathbf{x}_t, \mathbf{\Psi}\right) = \prod_{i=1}^{K} \prod_{j=1}^{K} p\left(s_t^{(i)} | s_{t-1}^{(j)}, \mathbf{x}_t, \mathbf{\Psi}\right)^{s_t^{(i)} s_{t-1}^{(j)}}, \tag{2}$$

where $\mathbf{\Psi}$ is a set of parameters of the transition model. The model allows state-transitions to be dependent on an external input $\mathbf{x}_t$— this can e.g. be used to model state-transitions caused by stimulation of subcortical structures involved in controlling cortical states [20]. Moving beyond this standard input output HMM formulation [21], we introduce hierarchically organised auxiliary latent variables $\mathbf{z}_t$ which represent the current state $\mathbf{s}_t$ through a binary tree. Using HME terminology, we refer to the nodes representing $\mathbf{z}_t$ as 'gates'. Each of the $K$ leaves of the tree (or, equivalently, each path through the tree) corresponds to one of the $K$ entries of $\mathbf{s}_t$ and we can thus represent $\mathbf{s}_t$ in the form

$$s_t^{(k)} = \prod_{l=1}^{L} \left(z_t^{(l)}\right)^{A_L^{(l,k)}} \left(1 - z_t^{(l)}\right)^{A_R^{(l,k)}}, \tag{3}$$

where $A_L$ and $A_R$ are adjacency matrices which indicate whether state $k$ is in the left or right branch of gate $l$, respectively (see [19]). Using this representation, $\mathbf{s}_t$ is deterministic given $\mathbf{z}_t$ which significantly simplifies the inference process. The auxiliary latent variables $z_t^{(l)}$ are Bernoulli random variables and we chose their conditional probability distribution to be

$$p(z_t^{(l)} = 1 | \mathbf{x}_t^{(l)}, \mathbf{s}_{t-1}, \mathbf{v}_l) = \sigma\left(\mathbf{v}_l^\top \mathbf{u}_t^{(l)}\right). \tag{4}$$

Here, $\sigma(\cdot)$ is the logistic sigmoid, $\mathbf{v}_l$ are the parameters of the $l$-th gate and $\mathbf{u}_t$ represents a concatenation of the previous state $\mathbf{s}_{t-1}$, the input $\mathbf{x}_t$ (which could for example represent population firing rate, time in trial or an external stimulus) and a constant term of unit value to model the prior probability of $z_0^{(l)} = 1$. This parametrisation significantly reduces the number of parameters used for the transition probabilities as compared to [10]. To enforce stronger temporal locality and less jumping between states we could also reduce this probability to be conditioned only on previous activations of a sub-tree of the HMDT instead of all population states.

## 2.2 Learning & Inference

For posterior inference over the model parameters we would need to infer the joint distribution over all stochastic variables conditioned on $\mathbf{X}$,

$$p\left(\mathbf{Y}, \mathbf{S}, \boldsymbol{\Phi}, \boldsymbol{\Psi}, \boldsymbol{\lambda}, \boldsymbol{\nu} | \mathbf{X}\right) = p\left(\mathbf{Y}|\mathbf{S}, \mathbf{X}, \boldsymbol{\Phi}\right) p\left(\mathbf{S}|\mathbf{X}, \boldsymbol{\Psi}\right) p\left(\boldsymbol{\Phi}|\boldsymbol{\lambda}\right) p\left(\boldsymbol{\lambda}\right) p\left(\boldsymbol{\Psi}|\boldsymbol{\nu}\right) p\left(\boldsymbol{\nu}\right) \tag{5}$$

where $\mathbf{Y}$ is the set of $\mathbf{y}_t$'s, $\boldsymbol{\Phi}$ and $\boldsymbol{\Psi}$ are the sets of parameters for the emission and gating distributions, respectively, and $\boldsymbol{\lambda}$ and $\boldsymbol{\nu}$ are the hyperparameters for the parameter priors. Since there is no closed form solution for this distribution, we use a variational approximation [8]. We assume that the posterior factorises as

$$q\left(\mathbf{S}, \boldsymbol{\Phi}, \boldsymbol{\Psi}, \boldsymbol{\lambda}, \boldsymbol{\nu}\right) = q\left(\mathbf{S}\right) q\left(\boldsymbol{\Phi}\right) q\left(\boldsymbol{\Psi}\right) q\left(\boldsymbol{\lambda}\right) q\left(\boldsymbol{\nu}\right) \tag{6}$$

$$= q\left(\mathbf{S}\right) \prod_{k=1}^{K} \prod_{c=1}^{C} q\left(\boldsymbol{\phi}_k^{(c)}\right) q\left(\boldsymbol{\lambda}_k^{(c)}\right) \prod_{l=1}^{L} q\left(\boldsymbol{\psi}_l\right) q\left(\boldsymbol{\nu}_l\right), \tag{7}$$

and find the variational approximation to the posterior over parameters, $q\left(\mathbf{S}, \boldsymbol{\Phi}, \boldsymbol{\Psi}, \boldsymbol{\lambda}, \boldsymbol{\nu}\right)$, by optimising the variational lower bound $\mathcal{L}(q)$ to the evidence

$$\mathcal{L}(q) := \sum_{\mathbf{S}} \iiiint q\left(\mathbf{S}, \boldsymbol{\Phi}, \boldsymbol{\Psi}, \boldsymbol{\lambda}, \boldsymbol{\nu}\right) \ln \frac{p\left(\mathbf{Y}, \mathbf{S}, \boldsymbol{\Phi}, \boldsymbol{\Psi}, \boldsymbol{\lambda}, \boldsymbol{\nu} | \mathbf{X}\right)}{q\left(\mathbf{S}, \boldsymbol{\Phi}, \boldsymbol{\Psi}, \boldsymbol{\lambda}, \boldsymbol{\nu}\right)} \mathrm{d}\boldsymbol{\Phi} \mathrm{d}\boldsymbol{\Psi} \mathrm{d}\boldsymbol{\lambda} \mathrm{d}\boldsymbol{\nu} \tag{8}$$

$$\leq \ln \sum_{\mathbf{S}} \iiiint p\left(\mathbf{Y}, \mathbf{S}, \boldsymbol{\Phi}, \boldsymbol{\Psi}, \boldsymbol{\lambda}, \boldsymbol{\nu} | \mathbf{X}\right) \mathrm{d}\boldsymbol{\Phi} \mathrm{d}\boldsymbol{\Psi} \mathrm{d}\boldsymbol{\lambda} \mathrm{d}\boldsymbol{\nu} = \ln p\left(\mathbf{Y}|\mathbf{X}\right). \tag{9}$$

We use variational Expectation-Maximisation (VBEM) to perform alternating updates on the posterior over latent state variables and the posterior over model parameters. To infer the posterior over latent variables (i.e. responsibilities), we use a modified forward-backward algorithm as proposed in [22] (see also [8]). In order to perform the forward and backward steps, they propose the use of subnormalised probabilities of the form

$$\tilde{p}\left(s_t^{(i)}|s_{t-1}^{(j)}, \mathbf{x}_t, \boldsymbol{\Psi}\right) := \exp\left(\mathbb{E}_{\boldsymbol{\Psi}}\left[\ln p\left(s_t^{(i)}|s_{t-1}^{(j)}, \mathbf{x}_t, \boldsymbol{\Psi}\right)\right]\right) \tag{10}$$

$$\tilde{p}\left(y_t|\mathbf{x}_t, \boldsymbol{\Phi}_i\right) := \exp\left(\mathbb{E}_{\boldsymbol{\Phi}_i}\left[\ln p\left(\mathbf{y}_t|\mathbf{x}_t, \boldsymbol{\Phi}_i\right)\right]\right) \tag{11}$$

for the state-transition probabilities and emission probabilities. Since all relevant probabilities in our model are over discrete variables, it would be straightforward to normalise those probabilities, but we found that normalisation did not noticeably change results.

With the approximations from above, the forward probability can thus be written as

$$\alpha\left(s_t^{(i)}\right) = \frac{1}{\tilde{C}_t} \tilde{p}\left(\mathbf{y}_t|s_t^{(i)}, \mathbf{x}_t, \boldsymbol{\phi}\right) \sum_{j=1}^{K} \alpha\left(s_{t-1}^{(j)}\right) \tilde{p}\left(s_t^{(i)}|s_{t-1}^{(j)}, \mathbf{x}_t, \boldsymbol{\Psi}\right), \tag{12}$$

where $\alpha(s_t^{(i)})$ is the probability-mass of state $s_t^{(i)}$ given previous time steps and $\tilde{C}_t$ is a normalisation constant. Similar to the forward step, the backward recursion takes the form

$$\beta\left(s_t^{(i)}\right) = \frac{1}{\tilde{C}_t} \sum_{j=1}^{K} \beta_t\left(s_{t+1}^{(j)}\right) \tilde{p}\left(\mathbf{y}_{t+1}|s_{t+1}^{(j)}, \mathbf{x}_{t+1}, \boldsymbol{\phi}\right) \tilde{p}\left(s_{t+1}^{(j)}|s_t^{(i)}, \mathbf{x}_t, \boldsymbol{\Psi}\right). \qquad (13)$$

Using the forward and backward equation steps we can infer the state posteriors [8]. Given the state posteriors, the logarithm of the approximate parameter posterior for each of the nodes takes the form

$$\ln q^{\star}\left(\boldsymbol{\omega}_n\right) = \sum_{t=1}^{T} \eta_t^{(n)} \ln p\left(\mu_t^{(n)}|\mathbf{x}_t^{(n)}, \boldsymbol{\omega}_n, (\dots)\right) + \mathbb{E}_{\boldsymbol{\gamma}_n}\left[\ln p\left(\boldsymbol{\omega}_n|\boldsymbol{\gamma}_n\right)\right] + \text{const.} \qquad (14)$$

where $\boldsymbol{\omega}_n$ are the parameters of the $n$-th node and $p\left(\boldsymbol{\omega}_n|\boldsymbol{\gamma}_n\right)$ is the prior over the parameters. Here, $\eta_t^{(n)}$ is the posterior responsibility or estimated influence of node $n$ on the $t^{\text{th}}$ observation and $\mu_t^{(n)}$ denotes the expected output (known for state nodes) of node $n$ (see supplement for details). This equation also holds for a tree structure with multinomial gates and for non-binary emission models such as Poisson and linear models. The above equations are valid for maximum likelihood inference, except that all parameter priors are removed, and the expectations of log-likelihoods reduce to log-likelihoods We use logistic regression for all emission probabilities and gates, and a local variational approximation to the logistic sigmoid as presented in [18].

As parameter priors we use anisotropic Gaussians with individual Gamma priors on each diagonal entry of the precision matrix. With this prior structure we can perform automatic relevance determination [23]. We chose shape parameter $a_0 = 1 \times 10^{-2}$ and rate parameter $b_0 = 1 \times 10^{-4}$, leading to a broad Gamma hyperprior [19]. In many applications, it will be reasonable to assume that neurons in close-by states of the tree show similar response characteristics (similar parameters). The hierarchical organisation of the model yields a natural structure for hierarchical priors which can encourage parameter similarity[2].

## 2.3 Details of simulated and neurophysiological data

To assess and illustrate our model, we simulated a population recording with trials of $3\,\text{s}$ length (20 neurons, $10\,\text{ms}$ time bins). As illustrated in Fig. 1 A, we modelled one low-firing-rate down state (*down*, base firing rate $0.5\,\text{Hz}$) and two up states (*up-low* and *up-high*, with base firing rates of 5, and $50\,\text{Hz}$ respectively). The root node switched between up and down states, whereas a second node controlled transitions between the two types of up-states. *Up-high* states only occurred transiently, modelling synchronised bouts of activity. In the down state, neurons have a $10\,\text{ms}$ refractory period, during up states they exhibit bursting activity. Transitions from *down* to *up* go mainly via *up-high* to *up-low*, while down-transitions go from *up-low* to *down*; stimulation increases the probability of being in one of the up states. A pulse-stimulus occurred at time $1\,\text{s}$ of each trial. Each model was fit on a set of 20 trials and evaluated on a different test set of 20 trials. For each training set, 24 random parameter initialisations were drawn and the one with highest evidence was chosen for evaluation. State predictions were evaluated using the Viterbi algorithm [24, Ch. 13].

We analysed a recording from visual cortex (V1) of an anaesthetised macaque [2]. The data-set consisted of 1600 presentations of drifting gratings (16 directions, 100 trials each), each lasting $2\,\text{s}$. Experimental details are described in [2]. For each trial, we kept a segment of $500\,\text{ms}$ before and after a stimulus presentation, resulting in trials of length $3\,\text{s}$ each. We binned and binarised spike trains in $50\,\text{ms}$ bins. Additional spikes (present in $(5.45 \pm 1.56)\,\%$ of bins) were discarded by the binarisation procedure. We chose the representation of the stimulus to be the outer product of the two vectors $[1, \sin(\vartheta), \cos(\vartheta)]$, where $\vartheta$ is the phase of the grating, and $[1, \sin(\theta), \cos(\theta), \sin(2\theta), \cos(2\theta)]$ for the direction $\theta$ of the grating. This resulted in a 15 dimensional stimulus-parametrisation, and made it possible to represent tuning-curves with orientation and direction selectivity, as well as modulation of firing rates by stimulus phase. The only gate input was chosen to be an indicator function with unit value during stimulus presentation and zero value otherwise. Post-spike filters were parametrised using five cubic b-splines for the last 10 bins with a bin width of $50\,\text{ms}$.

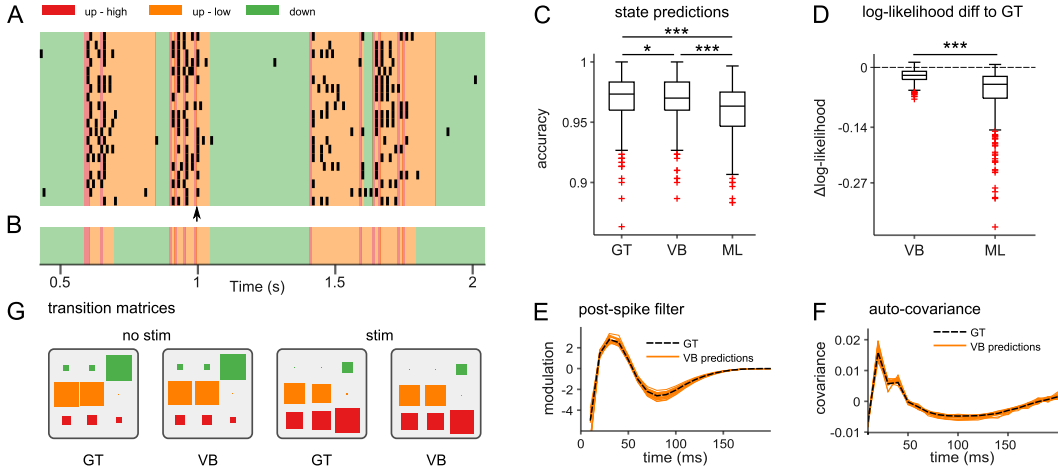

Figure 2: **Performance of the model on simulated data. A)** Example rasters sampled using ground truth (GT) parameters, colors indicate sequence of underlying population states. **B)** For the sample from (**A**), the state-sequence decoded with our variational Bayes (VB) method matches the decoded sequence using GT parameters. **C)** Comparison of state-decoding performance using GT parameters, VB and maximum likelihood (ML) learning (Wilcoxon ranksum, * $p < 0.05$; *** $p \ll 0.001$). **D)** Model performance quantified using per-data-point log-likelihood difference between estimated and GT-model on test-set. Our VB method outperforms ML (Wilcoxon ranksum, *** $p \ll 0.001$), and both models considerably outperform a 1-state GLM (not shown). **E)** Estimated post-spike filters match the GT values well (depicted are the filters from one of the cross-validated models). **F)** Comparison of the autocorrelation of the ground truth data and samples drawn from the VB fit as in (**E**). **G)** GT (top) and VB estimated (bottom) transition matrices in absence (left) or presence (right) of a stimulus.

## 3 Results

### 3.1 Results on simulated data

To illustrate our model and to evaluate the estimation procedure on data with known ground truth, we used a simulated population recording of 20 neurons by sampling from our model (details in Methods, see Fig. 2 A). In this simulation, the up-state had much higher firing rates than the down-state. It was therefore possible to decode the underlying states from the population spike trains with high accuracy (Fig. 2 B). For the VB method, we used the posterior mean over parameters for state-inference. In addition, we compared both of these approaches to state-decoding based on a model estimated using maximum likelihood learning. All three models showed similar performance, but the decoding advantage of the 3-state VB model was statistically significant (using pairwise comparisons, Fig. 2 C).

We also directly evaluated performance of the VB and ML methods for parameter estimation by calculating the log-likelihood of the data on held-out test-data, and found that our VB method performed significantly better than the ML method (Fig. 2 D). Finally, we also compared the estimated post-spike filters (Fig. 2 E), auto-correlation functions (Fig. 2 F) and state-transition matrices (Fig. 2 G) and found an excellent agreement between the GT parameters and the estimates returned by VB.

To test whether the VB method is able to determine the correct model complexity, we fit an over-parameterised model with 3 layers and potentially 8 states to the simulation data. The best model fit from 200 random restarts (lower bound of $-2.24 \times 10^4$, no cross-validation, results not shown) only used 3 out of the 8 possible states (the other 5 states had a probability of less than $0.5\,\%$). Therefore, in this example, the best lower bound is achieved by a model with correct, and low, complexity.

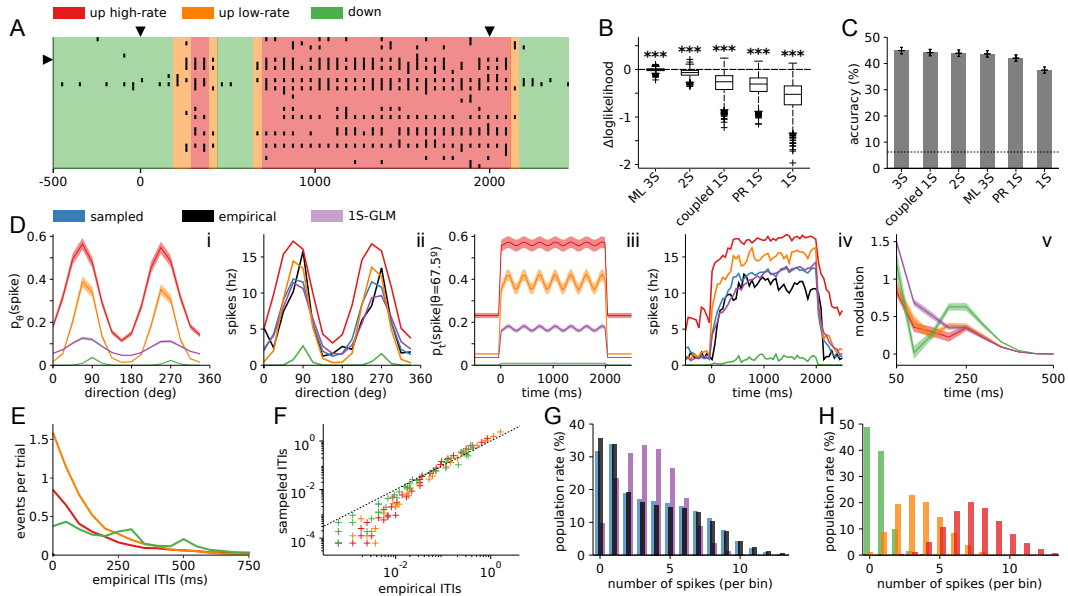

Figure 3: **Results for population recordings from V**1. **A)** Raster plot of population response to a drifting grating with orientation 67.5°. Arrows indicate stimulus onset and offset, colours show the most likely state sequence inferred with the 3-state variational Bayes (3S-VB) model. **B)** Cross-validated log-likelihoods per trial, relative to the 3S-VB model. **C)** Stimulus decoding performance, in percentage of correctly decoded stimuli (16 discrete stimuli, chance level 6.25 %), using maximum-likelihood decoding.
**D)** Tuning properties of an example neuron. i) Orientation tuning calculated from the tuning-parameters of 3S-VB (red, orange, green) or 1-state GLM (purple). iii) Temporal component of tuning parameters. ii) Orientation tuning measured from sampled data of the estimated model, each line representing one state. Note that the firing rate also depends on state-transitions and post-spike filters. iv) Peri-stimulus time-histograms (PSTHs) estimated from samples of the estimated models. v) Post-spike filters for each state, and comparison with 1-state GLM (purple). **E)** Distributions of times spent in each state, i.e. inter-transition intervals (ITIs), estimated from the empirical data using 3S-VB. **F)** Comparison between distribution of ITIs in samples from model 3S-VB and in the Viterbi-decoded path (from E).
**G)** Histogram of population rates (i.e. number of synchronous spikes across the population in each 50 ms bin) for 3S-VB (blue), 1S (purple), and data (gray). **H)** Histograms of population rate for each state.

## 3.2 Results on neurophysiological recordings

We analysed a neural population recording from V1 to determine whether we could successfully identify cortical states by decoding the activity of the neural population, and whether accounting for state-dependence resulted in a more accurate statistical model of neural firing. While neurons generally responded robustly to the stimulus (3 D), firing rates were strongly modulated by internal states [2] (Fig. 3 A). We fit different models to data, and found that our 3-state model estimated with VB resulted in better cross-validation performance than either the 3-state model estimated with ML, the 2-state model or a 1-state GLM (i.e. a GLM without cross-neural couplings, Fig. 3 B). In addition we fit a fully coupled GLM (with cross-history terms as in [11, 13]), as well as one in which the total population count was used as a history feature using VB. These models were intermediate between the 1-state GLM and the 2-state model, i.e. both worse than the 3-state one. A 'flat' 3-states model with a single multinomial gate estimated with ML performed similarly to the hierarchical 3S-ML model. This is to be expected, as any differences in expressive power between the two models will only become substantial for a different choice of $\mathbf{x}_t$ or larger models.

We also evaluated the ability of different models to decode the stimulus, (i.e. the direction of the presented grating) from population spike trains. We evaluated the likelihood of each population spike train for each of the 16 stimulus directions, and decoded the stimulus which yielded the highest likelihood. The 3-state VB model shows best decoding performance among all tested models (3 C), and all models with state-dependence (3-state VB, 3-state ML, 2-state) outperformed the 1-state GLM. We sampled from the estimated 3S-VB model to evaluate to what extent the model captures the tuning properties of neurons (Fig. 3 D(ii & iv)). The example neuron shows strong modulation of base firing rate dependent on the population state, but not a qualitative change of the tuning properties (Fig. 3 D i-iv). The down-state post-spike filter (Fig. 3 D v) exhibits a small oscillatory component which is not present in the post-spike filters of the other states or the 1-state GLM.

Investigation of inter-transition-interval (ITI) distributions from the data (after Viterbi-decoding) shows heavy tails (Fig. 3 E). Comparison of ITI-distribution estimated from the empirical data and from sampled data (3S-VB) show good agreement, apart from small deficiencies of the model to capture the heavy tails of the empirical ITI distribution (Fig. 3 F). Finally, population rates (i.e. total number of spikes across the population) are often used as a summary-measure for characterizing cortical states [6]. We found that the distribution of population rates in the data was well matched by the distribution estimated from our model (Fig. 3 G) with the three states having markedly different population rate distributions (Fig. 3 H). Although a 1-state GLM also captured the tuning-properties of this neuron (Fig. 3 D) it failed to recover the distribution of population rates (Fig. 3 G).

# 4   Discussion

We presented a statistical method for extracting cortical states from multi-cell recordings of spiking activity. Our model is based on a 'state-dependent' GLM [10] in which the states are organised hierarchically and evolve over time according to a hidden Markov model. Whether, and in which situations, the best descriptions of cortical states are multi-dimensional, discrete or continuous [25, 2] is an open question [6], and models like the one presented here will help shed light on these questions. We showed that the use of variational inference methods makes it possible to estimate the posterior over parameters. Bayesian inference provides better model performance on limited data [13], uncertainty information, and is also an important building block for active learning approaches [14]. Finally, it can be used to determine the best model complexity: For example, one could start inference with a model containing only one state and iteratively add states (as in divisive clustering) until the variational bound stops increasing.

Cortical states can have a substantial impact on the firing and coding properties of cortical neurons [6] and interact with inter-area communication [4, 3]. Therefore, a better understanding of the interplay between cortical states and sensory information, and the role of cortical states in gating information in local cortical circuits will be indispensable for our understanding of how populations of neurons collectively process information. Advances in experimental technology enable us to record neural activity in large populations of neurons distributed across brain areas. This makes it possible to empirically study how cortical states vary across the brain, to identify pathways which influence state, and ultimately to understand their role in neural coding and computation. The combination of such data with statistical methods for identifying the organisation of cortical states holds great promise for making progress on understanding state-dependent information processing in the brain.

**Acknowledgements**

We are grateful to the authors of [2] for sharing their data (toliaslab.org/publications/ecker-et-al-2014/) and to Alexander Ecker, William McGhee, Marcel Nonnenmacher and David Janssen for comments on the manuscript. This work was funded by the German Federal Ministry of Education and Research (BMBF; FKZ: 01GQ1002, Bernstein Center Tübingen) and the Max Planck Society. Supplementary details and code are available at www.mackelab.org.

## Footnotes

[1]All derivations below can be generalised to model the emission probabilities by any kind of generalised linear model.

[2]See supplement for an example of how this could be implemented with Gaussian priors.

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
