[Supplementary Material]

# Supplement for:
# A Bayesian model for identifying hierarchically organised states in neural population activity

**Patrick Putzky**[1,2,3]**, Florian Franzen**[1,2,3]**, Giacomo Bassetto**[1,3]**, Jakob H. Macke**[1,3]

[1]Max Planck Institute for Biological Cybernetics, Tübingen
[2]Graduate Training Centre of Neuroscience, University of Tübingen
[3]Bernstein Center for Computational Neuroscience, Tübingen
patrick.putzky@gmail.com, florian.franzen@tuebingen.mpg.de
giacomo.bassetto@tuebingen.mpg.de, jakob@tuebingen.mpg.de

## Abstract

This document is a supplement to the paper [1], and includes more detailed derivations of relevant results. This supplement was not peer-reviewed. Code implementing the methods described in this paper will be available at www.mackelab.org.s

## Contents

# 1 HMM formulation

## 1.1 Definition of variables

- K - number of states
- L - number of gates (auxiliary latent variables) with L = K - 1
- T - number of time steps
- C - total number of cells
- $\mathbf{s}$ - state variable, 1-of-K vector
- $\mathbf{y}$ - target variable, C x 1 binary vector which indicates whether a cell spiked (1) or not (0)
- $\mathbf{x}$ - vector of covariates (such as external stimulation, firing history)
- $\mathbf{\Psi}$ - gating parameters
- $\mathbf{\Phi}$ - regression parameters

## 1.2 Model description

The standard representation of a first order input-output hidden Markov model [2, 3] can be written as

$$p\left(\mathbf{S}_{0:T-1}, \mathbf{Y}_{0:T-1} | \mathbf{X}_{0:T-1}\right) = p\left(\mathbf{s}_0 | \mathbf{x}_0\right) p\left(\mathbf{y}_0 | \mathbf{s}_0, \mathbf{x}_0\right) \prod_{t=1}^{T-1} p\left(\mathbf{s}_t | \mathbf{s}_{t-1}, \mathbf{x}_t\right) p\left(\mathbf{y}_t | \mathbf{s}_t, \mathbf{x}_t\right). \tag{1}$$

The current state $\mathbf{s}_t$ at time $t$ depends on the previous state $\mathbf{s}_{t-1}$ and some input $\mathbf{x}_t$. Specific to the input-output hidden Markov model (IOHMM) compared to a homogeneous HMM is the fact that state transitions and state emissions can be conditioned on some external input $\mathbf{x}_t$ as well.

We write the transition probabilities of the IOHMM as

$$p\left(\mathbf{s}_t | \mathbf{s}_{t-1}, \mathbf{x}_t, \mathbf{\Psi}\right) = \prod_{i=1}^{K} \prod_{j=1}^{K} p\left(s_t^{(i)} | s_{t-1}^{(j)}, \mathbf{x}_t, \mathbf{\Psi}\right)^{s_t^{(i)} s_{t-1}^{(j)}} \tag{2}$$

The external input $\mathbf{x}_t$ can e.g. consist of features representing a physical stimulus or the filtered populating spiking activity as in [4].

Similarly, the emission probabilities are represented by conditioning on the current state $\mathbf{s}_t$ and some input $\mathbf{x}_t$. Note here that the emission probabilities between neurons are assumed to be conditionally independent given the current population state.

$$p\left(\mathbf{y}_t | \mathbf{x}_t, \mathbf{s}_t, \mathbf{\Phi}\right) = \prod_{i=1}^{K} \prod_{c=1}^{C} p\left(y_t^{(c)} | \mathbf{x}_t^{(c)}, \phi_i^{(c)}\right)^{s_t^{(i)}} \tag{3}$$

The covariates $\mathbf{x}_t^{(c)}$ can contain features of the external stimulus, the neurons' own spiking history or the spike history of other neurons in the population [5, 6].

In order to represent a hierarchical organisation of population states, we introduce hierarchically organised auxiliary latent variables which represent the current state $\mathbf{s}_t$. Here, $\mathbf{s}_t$ is a 1-of-K vector which represents the unique active path within the hierarchy of $z_t$'s (binary variables). We can represent $\mathbf{s}_t$ in the form

$$s_t^{(k)} = \prod_{l=1}^{L} \left(z_t^{(l)}\right)^{A_L^{(l,k)}} \left(1 - z_t^{(l)}\right)^{A_R^{(l,k)}} \tag{4}$$

Figure 1: **Graphical model of the hierarchical Markov decision process with four hidden states.** The hidden states $s_t$ are deterministic given the auxiliary latent variable $z_t$, which is organised in a hierarchical fashion.

where $A_L$ is a binary matrix with $A_L^{(l,i)} = 1$ if and only if the $i$-th state is in the left branch of gate $l$ and $A_R$ is a binary matrix with $A_R^{(l,i)} = 1$ if and only if the $i$-th state is in the right branch of gate $l$.

The conditional probability of the population being in state $\mathbf{s}_t^{(i)}$ can be rewritten as

$$p\left(s_t^{(i)}|s_{t-1}^{(j)}, \mathbf{x}_t, \mathbf{\Psi}\right) = \prod_{l=1}^{L} p\left(z_t^{(l)} = 1|s_{t-1}^{(j)}, \mathbf{x}_t^{(l)}, \boldsymbol{\psi}_l\right)^{A_L^{(l,i)}} p\left(z_t^{(l)} = 0|s_{t-1}^{(j)}, \mathbf{x}_t^{(l)}, \boldsymbol{\psi}_l\right)^{A_R^{(l,i)}} \quad (5)$$

where $p\left(z_t^{(l)}|s_{t-1}^{(j)}, \mathbf{x}_t^{(l)}, \boldsymbol{\psi}_l\right)$ follows a Bernoulli distribution such that

$$p\left(z_t^{(l)}|s_{t-1}^{(j)}, \mathbf{x}_t^{(l)}, \boldsymbol{\psi}_l\right) = g\left(s_{t-1}^{(j)}, \mathbf{x}_t^{(l)}, \boldsymbol{\psi}_l\right)^{z_t^{(l)}} \left(1 - g\left(s_{t-1}^{(j)}, \mathbf{x}_t^{(l)}, \boldsymbol{\psi}_l\right)\right)^{\left(1-z_t^{(l)}\right)}. \quad (6)$$

Here, $g$ is an activation function which maps into the interval $(0,1)$.

### 1.3 Variational distribution

We can now write the joint distribution over observed and latent variables - including the distribution over model parameters - as

$$\begin{aligned}
p\left(\mathbf{Y}, \mathbf{S}, \mathbf{\Phi}, \mathbf{\Psi}, \boldsymbol{\lambda}, \boldsymbol{\nu}|\mathbf{X}\right) &= p\left(\mathbf{Y}|\mathbf{S}, \mathbf{X}, \mathbf{\Phi}\right) p\left(\mathbf{S}|\mathbf{X}, \mathbf{\Psi}\right) p\left(\mathbf{\Phi}|\boldsymbol{\lambda}\right) p\left(\boldsymbol{\lambda}\right) p\left(\mathbf{\Psi}|\boldsymbol{\nu}\right) p\left(\boldsymbol{\nu}\right) \\
&= p\left(\mathbf{Y}|\mathbf{S}, \mathbf{X}, \mathbf{\Phi}\right) p\left(\mathbf{S}|\mathbf{X}, \mathbf{\Psi}\right) \prod_{k=1}^{K} p\left(\boldsymbol{\phi}_k|\boldsymbol{\lambda}_k\right) p\left(\boldsymbol{\lambda}_k\right) \prod_{l=1}^{L} p\left(\boldsymbol{\psi}_l|\boldsymbol{\nu}_l\right) p\left(\boldsymbol{\nu}_l\right).
\end{aligned}$$
$$(7)$$

Here $\mathbf{Y}$, is the set of $\mathbf{y}_t$'s, $\mathbf{\Phi}$ and $\mathbf{\Psi}$ are the sets of parameters for the emission and gating distributions, respectively, and $\boldsymbol{\lambda}$ and $\boldsymbol{\nu}$ are the hyperparameters for the parameter priors. Since there is no closed form solution for this distribution, we use a variational approximation [7]. We assume that the posterior factorises as

$$q\left(\mathbf{S}, \mathbf{\Phi}, \mathbf{\Psi}, \boldsymbol{\lambda}, \boldsymbol{\nu}\right) = q\left(\mathbf{S}\right) q\left(\mathbf{\Phi}\right) q\left(\mathbf{\Psi}\right) q\left(\boldsymbol{\lambda}\right) q\left(\boldsymbol{\nu}\right) \quad (8)$$

$$= q\left(\mathbf{S}\right) \prod_{k=1}^{K} \prod_{c=1}^{C} q\left(\boldsymbol{\phi}_k^{(c)}\right) q\left(\boldsymbol{\lambda}_k^{(c)}\right) \prod_{l=1}^{L} q\left(\boldsymbol{\psi}_l\right) q\left(\boldsymbol{\nu}_l\right), \quad (9)$$

and find the variational approximation to the posterior over parameters, $q\left(\mathbf{S}, \boldsymbol{\Phi}, \boldsymbol{\Psi}, \boldsymbol{\lambda}, \boldsymbol{\nu}\right)$, by optimising the variational lower bound $\mathcal{L}(q)$ on the evidence,

$$\mathcal{L}(q) := \sum_{\mathbf{S}} \iiiint q\left(\mathbf{S}, \boldsymbol{\Phi}, \boldsymbol{\Psi}, \boldsymbol{\lambda}, \boldsymbol{\nu}\right) \ln \frac{p\left(\mathbf{Y}, \mathbf{S}, \boldsymbol{\Phi}, \boldsymbol{\Psi}, \boldsymbol{\lambda}, \boldsymbol{\nu} | \mathbf{X}\right)}{q\left(\mathbf{S}, \boldsymbol{\Phi}, \boldsymbol{\Psi}, \boldsymbol{\lambda}, \boldsymbol{\nu}\right)} \mathrm{d}\boldsymbol{\Phi} \mathrm{d}\boldsymbol{\Psi} \mathrm{d}\boldsymbol{\lambda} \mathrm{d}\boldsymbol{\nu} \tag{10}$$

$$\leq \ln \sum_{\mathbf{S}} \iiiint p\left(\mathbf{Y}, \mathbf{S}, \boldsymbol{\Phi}, \boldsymbol{\Psi}, \boldsymbol{\lambda}, \boldsymbol{\nu} | \mathbf{X}\right) \mathrm{d}\boldsymbol{\Phi} \mathrm{d}\boldsymbol{\Psi} \mathrm{d}\boldsymbol{\lambda} \mathrm{d}\boldsymbol{\nu} = \ln p\left(\mathbf{Y} | \mathbf{X}\right). \tag{11}$$

We further assume that the approximate posterior is fully factorised:

$$
\begin{aligned}
q\left(\mathbf{S}, \boldsymbol{\Phi}, \boldsymbol{\Psi}, \boldsymbol{\lambda}, \boldsymbol{\nu}\right) = & q\left(\mathbf{S}\right) q\left(\boldsymbol{\Phi}\right) q\left(\boldsymbol{\Psi}\right) q\left(\boldsymbol{\lambda}\right) q\left(\boldsymbol{\nu}\right) \\
= & q\left(\mathbf{S}\right) \prod_{k=1}^{K} \prod_{c=1}^{C} q\left(\boldsymbol{\phi}_k^{(c)}\right) q\left(\boldsymbol{\lambda}_k^{(c)}\right) \prod_{l=1}^{L} q\left(\boldsymbol{\psi}_l\right) q\left(\boldsymbol{\nu}_l\right)
\end{aligned} \tag{12}
$$

## 1.4 Variational Inference

We start by deriving the optimal factor of the latent variables $\ln q^\star\left(\mathbf{S}\right)$. This is done in the same fashion as in [3, Ch. 10.2.1], including the terminology used. The logarithm of the optimal factor is given by the equation

$$\ln q^\star\left(\mathbf{S}\right) = \mathbb{E}_{\boldsymbol{\Phi}, \boldsymbol{\Psi}, \boldsymbol{\lambda}, \boldsymbol{\nu}}\left[\ln p\left(\mathbf{t}, \mathbf{S}, \boldsymbol{\Phi}, \boldsymbol{\Psi}, \boldsymbol{\lambda}, \boldsymbol{\nu} | \mathbf{X}\right)\right]. \tag{13}$$

While deriving the optimal factor, we will ignore all terms that are not dependent on $\mathbf{S}$ for now by summarising them in an additive constant term such that

$$
\begin{aligned}
\ln q^\star\left(\mathbf{S}\right) & = \mathbb{E}_{\boldsymbol{\Phi}, \boldsymbol{\Psi}}\left[\ln p\left(\mathbf{t} | \mathbf{S}, \mathbf{X}, \boldsymbol{\Phi}\right) p\left(\mathbf{S} | \mathbf{X}, \boldsymbol{\Psi}\right)\right] - \tilde{\mathcal{Z}}\left(\mathbf{Y} | \mathbf{X}\right) \\
& = \mathbb{E}_{\boldsymbol{\Phi}}\left[\ln p\left(\mathbf{Y} | \mathbf{S}, \mathbf{X}, \boldsymbol{\Phi}\right)\right] + \mathbb{E}_{\boldsymbol{\Psi}}\left[\ln p\left(\mathbf{S} | \mathbf{X}, \boldsymbol{\Psi}\right)\right] - \tilde{\mathcal{Z}}\left(\mathbf{Y} | \mathbf{X}\right).
\end{aligned} \tag{14}
$$

Here, the expectation of each term is only taken with respect to variables that are included in the term. The normalisation constant $\tilde{\mathcal{Z}}\left(\mathbf{Y} | \mathbf{X}\right)$ will play a role for calculating the variational lower bound, as will be shown later [7]. It is estimated as a side product of the forward-backward algorithm. By substituting 5 and 3 into the right-hand side of 14. we find

$$
\begin{aligned}
\ln q^\star\left(\mathbf{S}\right) = & \sum_{t=0}^{T-1} \sum_{i=1}^{K} \sum_{j=1}^{K} s_t^{(i)} s_{t-1}^{(j)} \mathbb{E}_{\boldsymbol{\Psi}}\left[\ln p\left(s_t^{(i)} | s_{t-1}^{(j)}, \mathbf{x}_t, \boldsymbol{\Psi}\right)\right] \\
& + \sum_{t=0}^{T-1} \sum_{i=1}^{K} s_t^{(i)} \mathbb{E}_{\boldsymbol{\phi}_i}\left[\ln p\left(\mathbf{y}_t | \mathbf{X}_t, \boldsymbol{\phi}_i\right)\right] - \tilde{\mathcal{Z}}\left(\mathbf{Y} | \mathbf{X}\right).
\end{aligned} \tag{15}
$$

This term takes a similar form as in the maximum likelihood framework, which is appropriate to perform the forward-backward algorithms (see [8, 7]). Instead of log-probabilities however, we are dealing with expectations of log-probabilities. We therefore need to approximate the transition and emission probabilities.

We will approximate the transition and emission probabilities by (see [8, 7]):

$$\tilde{p}\left(s_t^{(i)} | s_{t-1}^{(j)}, \mathbf{x}_t, \boldsymbol{\Psi}\right) := \exp\left(\mathbb{E}_{\boldsymbol{\Psi}}\left[\ln p\left(s_t^{(i)} | s_{t-1}^{(j)}, \mathbf{x}_t, \boldsymbol{\Psi}\right)\right]\right); \sum_{i=1}^{K} \tilde{p}\left(s_t^{(i)} | s_{t-1}^{(j)}, \mathbf{x}_t, \boldsymbol{\Psi}\right) \leq 1 \tag{16}$$

for the transition probabilities and

$$\tilde{p}\left(\mathbf{y}_t | s_t^{(i)}, \mathbf{x}_t, \boldsymbol{\phi}\right) = \prod_{c=1}^{C} \tilde{p}\left(y_t^{(c)} | \mathbf{x}_t^{(c)}, \boldsymbol{\phi}_i^{(c)}\right) \tag{17}$$

with

$$\tilde{p}\left(y_t^{(c)}|\mathbf{x}_t^{(c)}, \boldsymbol{\phi}_i^{(c)}\right) := \exp\left(\mathbb{E}_{\boldsymbol{\phi}_i^{(c)}}\left[\ln p\left(y_t^{(c)}|\mathbf{x}_t^{(c)}, \boldsymbol{\phi}_i^{(c)}\right)\right]\right); \int \tilde{p}\left(y_t^{(c)}|\mathbf{x}_t^{(c)}, \boldsymbol{\phi}_i^{(c)}\right) \mathrm{d}y_t^{(c)} \le 1 \tag{18}$$

for the emission probabilities, where we have made use of the fact that $\mathbb{E}_{\boldsymbol{\phi}_i}\left[\ln p\left(\mathbf{y}_t|\mathbf{X}_t, \boldsymbol{\phi}_i\right)\right]$ factorises into independent emission factors for each cell. Note that those probabilities are sub-normalised (do not sum to 1) [8, 7]. Since all probabilities we are dealing with for the purpose of the main article are discrete it would be straightforward to normalise those probabilities. In fact, doing so lead to very similar results as presented in the main article. Here, we will however conform to [8, 7].

### 1.4.1 Forward-backward algorithm

We use the approximate transition and emission probabilities from above to calculate the forward pass with

$$\tilde{C}_t\alpha\left(s_t^{(i)}\right) = \tilde{\alpha}\left(s_t^{(i)}\right) = \tilde{p}\left(\mathbf{y}_t|s_t^{(i)}, \mathbf{x}_t, \boldsymbol{\phi}\right)\sum_{j=1}^{K}\alpha\left(s_{t-1}^{(j)}\right)\tilde{p}\left(s_t^{(i)}|s_{t-1}^{(j)}, \mathbf{x}_t, \boldsymbol{\Psi}\right) \tag{19}$$

where

$$\tilde{C}_t = \sum_{k=1}^{K}\tilde{\alpha}\left(s_t^{(k)}\right).$$

is the normalisation constant which ensures that $\alpha\left(s_t^{(i)}\right)$ sums to one.

Similarly, we can calculate the backward pass where we utilise the normalisation constants from the forward pass (see [3, Ch. 13.2]):

$$\beta\left(s_t^{(i)}\right) = \frac{1}{\tilde{C}_t}\sum_{j=1}^{K}\beta_t\left(s_{t+1}^{(j)}\right)\tilde{p}\left(\mathbf{y}_{t+1}|s_{t+1}^{(j)}, \mathbf{x}_{t+1}, \boldsymbol{\phi}\right)\tilde{p}\left(s_{t+1}^{(j)}|s_t^{(i)}, \mathbf{x}_t, \boldsymbol{\Psi}\right). \tag{20}$$

with $\beta\left(s_{T-1}^{(i)}\right) = 1$ as the starting condition.

A further use of the normalisation constants $\tilde{C}_t$ is for calculating the normalisation constant $\tilde{\mathcal{Z}}\left(\mathbf{Y}|\mathbf{X}\right)$ of the approximate posterior $q(\mathbf{S})$ such that

$$\tilde{\mathcal{Z}}\left(\mathbf{Y}|\mathbf{X}\right) = \tilde{p}\left(\mathbf{Y}|\mathbf{X}\right) = \prod_{t=1}^{T}\tilde{C}_t. \tag{21}$$

This quantity will be useful for calculating the variational lower bound later.

Using the forward-backward recursions we can calculate the marginal $\mathbb{E}\left[s_t^{(i)}\right]$ and joint probabilities $\mathbb{E}\left[\mathbf{s}_t^{(i)}, \mathbf{s}_{t-1}^{(j)}\right]$ - which we will also refer to as responsibilities - by

$$\mathbb{E}\left[s_t^{(i)}\right] = \alpha\left(s_t^{(i)}\right)\beta\left(s_t^{(i)}\right), \tag{22}$$

$$\mathbb{E}\left[\mathbf{s}_t^{(i)}, \mathbf{s}_{t-1}^{(j)}\right] = \frac{1}{Z_t}\alpha\left(s_{t-1}^{(j)}\right)\tilde{p}\left(s_t^{(i)}|s_{t-1}^{(j)}, \mathbf{x}_t, \boldsymbol{\Psi}\right)\tilde{p}\left(\mathbf{y}_t|s_t^{(i)}, \mathbf{x}_t, \boldsymbol{\phi}\right)\beta\left(s_t^{(i)}\right). \tag{23}$$

### 1.4.2 Posterior inference

**Emission parameters** We next find the form of the factor $q^\star\left(\phi_k\right)$ which approximates the posterior over the parameters of state $k$. Following the same approach as above by moving terms independent of $\phi_k$ into a constant term we find

$$
\begin{aligned}
\ln q^\star\left(\phi_k^{(c)}\right) =& \mathbb{E}_{\mathbf{S}}\left[\ln p\left(\mathbf{y}^{(c)}|\mathbf{X}^{(c)},\phi_k^{(c)}\right)\right] + \mathbb{E}_{\boldsymbol{\lambda}_k^{(c)}}\left[\ln p\left(\phi_k^{(c)}|\boldsymbol{\lambda}_k^{(c)}\right)\right] + \text{const.} \\
=& \sum_{t=1}^{T}\mathbb{E}\left[s_t^{(k)}\right]\ln p\left(y_t^{(c)}|\mathbf{x}_t^{(c)},\phi_k^{(c)}\right) + \mathbb{E}_{\boldsymbol{\lambda}_k^{(c)}}\left[\ln p\left(\phi_k^{(c)}|\boldsymbol{\lambda}_k^{(c)}\right)\right] + \text{const.}
\end{aligned}
\tag{24}
$$

It can be seen that the log of factor $q^\star\left(\phi_k^{(c)}\right)$ is composed of the weighted log-likelihood of the data plus the expectation of the model parameters with respect to $q^\star\left(\boldsymbol{\lambda}_k^{(c)}\right)$. That means the more data is observed, the less influence does the prior have on the approximate posterior. Note that the log-likelihoods are weighted by the responsibilities of the respective state as described above.

The approximate posterior over the hyper-parameters for state $k$ can be written as

$$
\ln q^\star\left(\boldsymbol{\lambda}_k^{(c)}\right) = \mathbb{E}_{\phi_k^{(c)}}\left[\ln p\left(\phi_k^{(c)}|\boldsymbol{\lambda}_k^{(c)}\right)\right] + \ln p\left(\boldsymbol{\lambda}_k^{(c)}\right) + \text{const.}
\tag{25}
$$

**Gating parameters** We now turn to the approximate posterior over gating parameters. We will start with the posterior over all gating parameters $\boldsymbol{\Psi}$ and show that it naturally factorises into approximate posteriors for each gating node such that $\ln q^\star\left(\boldsymbol{\Psi}\right) = \sum_{l=1}^{L}\ln q^\star\left(\boldsymbol{\psi}_l\right)$. The logarithm of the factor $\ln q^\star\left(\boldsymbol{\Psi}\right)$ can be written as

$$
\ln q^\star\left(\boldsymbol{\Psi}\right) = \mathbb{E}_{\mathbf{S}}\left[\ln p\left(\mathbf{S}|\mathbf{X},\boldsymbol{\Psi}\right)\right] + \mathbb{E}_{\boldsymbol{\nu}}\left[\ln p\left(\boldsymbol{\Psi}|\boldsymbol{\nu}\right)\right] + \text{const.}
\tag{26}
$$

But as $p\left(\mathbf{S}|\mathbf{X},\boldsymbol{\Psi}\right)$ is defined over the $K$ states, some terms have to be rearranged to show that in fact $\ln q^\star\left(\boldsymbol{\Psi}\right)$ can be factorised into $\sum_{l=1}^{L}\ln q^\star\left(\boldsymbol{\psi}_l\right)$. By substituting the definitions for the conditional distributions of the equation above we find

$$
\ln q^\star\left(\boldsymbol{\Psi}\right) = \sum_{t=1}^{T}\sum_{i=1}^{K}\sum_{j=1}^{K}\mathbb{E}\left[\mathbf{s}_t^{(i)},\mathbf{s}_{t-1}^{(j)}\right]\ln p\left(s_t^{(i)}|s_{t-1}^{(j)},\mathbf{X}_t,\boldsymbol{\Psi}\right) + \sum_{l=1}^{L}\mathbb{E}_{\boldsymbol{\nu}_l}\left[\ln p\left(\boldsymbol{\psi}_l|\boldsymbol{\nu}_l\right)\right] + \text{const.}
\tag{27}
$$

where it can be seen that the second part of the equation already factorises into $L$ terms due to the model description. We will inspect the first part of the equation which is $\mathbb{E}_{\mathbf{S}}\left[\ln p\left(\mathbf{S}|\mathbf{X},\boldsymbol{\Psi}\right)\right]$ more closely to show that it also factorises into $L$ terms:

$$
\begin{aligned}
\mathbb{E}_{\mathbf{S}}\left[\ln p\left(\mathbf{S}|\mathbf{X},\boldsymbol{\Psi}\right)\right] =& \sum_{t=1}^{T}\sum_{i=1}^{K}\sum_{j=1}^{K}\mathbb{E}\left[\mathbf{s}_t^{(i)},\mathbf{s}_{t-1}^{(j)}\right]\ln p\left(s_t^{(i)}|s_{t-1}^{(j)},\mathbf{X}_t,\boldsymbol{\Psi}\right) \\
=& \sum_{t=1}^{T}\sum_{i=1}^{K}\sum_{j=1}^{K}\mathbb{E}\left[\mathbf{s}_t^{(i)},\mathbf{s}_{t-1}^{(j)}\right]\sum_{l=1}^{L}A_L^{(l,i)}\ln p\left(z_t^{(l)}=1|s_{t-1}^{(j)},\mathbf{x}_t^{(l)},\boldsymbol{\psi}_l\right) \\
& + A_R^{(l,i)}\ln p\left(z_t^{(l)}=0|s_{t-1}^{(j)},\mathbf{x}_t^{(l)},\boldsymbol{\psi}_l\right) \\
=& \sum_{l=1}^{L}\sum_{t=1}^{T}\sum_{i=1}^{K}\sum_{j=1}^{K}\mathbb{E}\left[\mathbf{s}_t^{(i)},\mathbf{s}_{t-1}^{(j)}\right]A_L^{(l,i)}\ln p\left(z_t^{(l)}=1|s_{t-1}^{(j)},\mathbf{x}_t^{(l)},\boldsymbol{\psi}_l\right) \\
& + A_R^{(l,i)}\ln p\left(z_t^{(l)}=0|s_{t-1}^{(j)},\mathbf{x}_t^{(l)},\boldsymbol{\psi}_l\right),
\end{aligned}
\tag{28}
$$

where we have made use of the definition in 6 and rearranged the sums to show that the equation factorises into $L$ independent terms.

Following this observation it is possible to derive the approximate posterior over the parameters for each gate separately with

$$
\begin{aligned}
\ln q^\star \left( \boldsymbol{\psi}_l \right) = & \sum_{t=1}^{T} \sum_{i=1}^{K} \sum_{j=1}^{K} \mathbb{E}\left[ \mathbf{s}_t^{(i)}, \mathbf{s}_{t-1}^{(j)} \right] A_L^{(l,i)} \ln p \left( z_t^{(l)} = 1 | s_{t-1}^{(j)}, \mathbf{x}_t^{(l)}, \boldsymbol{\psi}_l \right) \\
& + A_R^{(l,i)} \ln p \left( z_t^{(l)} = 0 | s_{t-1}^{(j)}, \mathbf{x}_t^{(l)}, \boldsymbol{\psi}_l \right) + \mathbb{E}_{\boldsymbol{\nu}_i} \left[ \ln p \left( \boldsymbol{\psi}_i | \boldsymbol{\nu}_i \right) \right] + \text{const.} \\
= & \sum_{t=1}^{T} \sum_{j=1}^{K} \ln p \left( z_t^{(l)} = 1 | s_{t-1}^{(j)}, \mathbf{x}_t^{(l)}, \boldsymbol{\psi}_l \right) \left( \sum_{i=1}^{K} \mathbb{E}\left[ \mathbf{s}_t^{(i)}, \mathbf{s}_{t-1}^{(j)} \right] A_L^{(l,i)} \right) \\
& \ln p \left( z_t^{(l)} = 0 | s_{t-1}^{(j)}, \mathbf{x}_t^{(l)}, \boldsymbol{\psi}_l \right) \left( \sum_{i=1}^{K} \mathbb{E}\left[ \mathbf{s}_t^{(i)}, \mathbf{s}_{t-1}^{(j)} \right] A_R^{(l,i)} \right) \\
& + \mathbb{E}_{\boldsymbol{\nu}_l} \left[ \ln p \left( \boldsymbol{\psi}_l | \boldsymbol{\nu}_l \right) \right] + \text{const.} \\
= & \sum_{t=1}^{T} \sum_{j=1}^{K} r_L^{(t,l)} \ln p \left( z_t^{(l)} = 1 | s_{t-1}^{(j)}, \mathbf{x}_t^{(l)}, \boldsymbol{\psi}_l \right) + r_R^{(t,l)} \ln p \left( z_t^{(l)} = 0 | s_{t-1}^{(j)}, \mathbf{x}_t^{(l)}, \boldsymbol{\psi}_l \right) \\
& + \mathbb{E}_{\boldsymbol{\nu}_l} \left[ \ln p \left( \boldsymbol{\psi}_l | \boldsymbol{\nu}_l \right) \right] + \text{const.}
\end{aligned}
\tag{29}
$$

where

$$
r_L^{(t,l,j)} = \sum_{i=1}^{K} \mathbb{E}\left[ \mathbf{s}_t^{(i)}, \mathbf{s}_{t-1}^{(j)} \right] A_L^{(l,i)} \qquad \text{and} \qquad r_R^{(t,l,j)} = \sum_{i=1}^{K} \mathbb{E}\left[ \mathbf{s}_t^{(i)}, \mathbf{s}_{t-1}^{(j)} \right] A_R^{(l,i)}
\tag{30}
$$

are the sums over the responsibilities of all states in the left subtree and in the right subtree of gate $l$, respectively. By normalising those quantities the equation can be rewritten as

$$
\begin{aligned}
\ln q^\star \left( \boldsymbol{\psi}_l \right) = & \sum_{j=1}^{K} \sum_{t=1}^{T} \mathbb{E}\left[ z_t^{pa(l)}, s_{t-1}^{(j)} \right] \ln p \left( \mathbb{E}\left[ z_t^{(l)} | z_t^{pa(l)}, s_{t-1}^{(j)} \right] | s_{t-1}^{(j)}, \mathbf{x}_t^{(l)}, \boldsymbol{\psi}_l \right) \\
& + \mathbb{E}_{\boldsymbol{\nu}_l} \left[ \ln p \left( \boldsymbol{\psi}_l | \boldsymbol{\nu}_l \right) \right] + \text{const.}
\end{aligned}
\tag{31}
$$

where

$$
\mathbb{E}\left[ z_t^{pa(l)}, s_{t-1}^{(j)} \right] = r_L^{(t,l,j)} + r_R^{(t,l,j)}
\tag{32}
$$

is the sum over the responsibilities of all states in the subtrees of gate $l$ which reflects the joint probability of the subtree of gate $l$ being active and the previous state being $s_j$, and

$$
\mathbb{E}\left[ z_t^{(l)} | z_t^{pa(l)}, s_{t-1}^{(j)} \right] = \frac{r_L^{(t,l,j)}}{\mathbb{E}\left[ z_t^{pa(l)}, s_{t-1}^{(j)} \right]}
\tag{33}
$$

are the normalised responsibilities from above. Here $\mathbb{E}\left[ z_t^{(l)} | z_t^{pa(l)}, s_{t-1}^{(j)} \right]$ represents the conditional probability of the left subtree of gate $l$ being active conditioned on the previous state being $s_j$ and gate $l$ being activated.

**General form**  Following these findings we can formalise a general form for all nodes, be it gating nodes or state nodes:

$$
\ln q^\star \left( \boldsymbol{\omega}_n \right) = \sum_{t=1}^{T} \eta_t^{(n)} \ln p \left( \mu_t^{(n)} | \mathbf{x}_t^{(n)}, \boldsymbol{\omega}_n, (\dots) \right) + \mathbb{E}_{\boldsymbol{\gamma}_n} \left[ \ln p \left( \boldsymbol{\omega}_n | \boldsymbol{\gamma}_n \right) \right] + \text{const.}
\tag{34}
$$

where $\boldsymbol{\omega}_n$ are the parameters of the $n$-th node and $p \left( \boldsymbol{\omega}_n | \boldsymbol{\gamma}_n \right)$ is the prior over the parameters. $\mu_t^{(n)}$ denotes the output of gate $n$ with $\mu_t^{(n)} = y_t^{(n)}$ for emission nodes and $\mu_t^{(n)} = \mathbb{E}\left[ z_t^{(n)} | z_t^{pa(n)}, s_{t-1}^{(j)} \right]$ for gate nodes. $\eta_t^{(n)}$ denotes the influence or responsibility of

node $n$ on observation $t$ with $\eta_t^{(n)} = \mathbb{E}\left[s_t^{(n)}\right]$ for state nodes and $\eta_t^{(n)} = \mathbb{E}\left[z_t^{pa(n)}, s_{t-1}^{(j)}\right]$ for gate nodes. This observation simplifies the derivations which we will perform later, since we can perform them on this general form. It should be noted that this equation holds for any kind of regression problem as long as the conditional probabilities for the gating nodes have the form as defined in 6.

## 1.5 Variational lower bound

For the HMDT we can evaluate the variational lower bound of the evidence. This bound is useful for evaluating model convergence and model comparison. As the lower bound must be monotonically increasing over iterations, a stopping criterion can be set by the amount of change of the lower bound. Furthermore, the bound is a good way of evaluating if all derivations have been performed and implemented correctly.

The lower bound of the hierarchical mixture of experts takes the form

$$
\begin{aligned}
\mathcal{L} =& \sum_{\mathbf{S}} \iiiint q\left(\mathbf{S}, \boldsymbol{\Phi}, \boldsymbol{\Psi}, \boldsymbol{\lambda}, \boldsymbol{\nu}\right) \ln \frac{p\left(\mathbf{Y}, \mathbf{S}, \boldsymbol{\Phi}, \boldsymbol{\Psi}, \boldsymbol{\lambda}, \boldsymbol{\nu}|\mathbf{X}\right)}{q\left(\mathbf{S}, \boldsymbol{\Phi}, \boldsymbol{\Psi}, \boldsymbol{\lambda}, \boldsymbol{\nu}\right)} \mathrm{d}\boldsymbol{\Phi}\mathrm{d}\boldsymbol{\Psi}\mathrm{d}\boldsymbol{\lambda}\mathrm{d}\boldsymbol{\nu} \\
=& \mathbb{E}\left[\ln p\left(\mathbf{Y}, \mathbf{S}, \boldsymbol{\Phi}, \boldsymbol{\Psi}, \boldsymbol{\lambda}, \boldsymbol{\nu}|\mathbf{X}\right)\right] - \mathbb{E}\left[\ln q\left(\mathbf{S}, \boldsymbol{\Phi}, \boldsymbol{\Psi}, \boldsymbol{\lambda}, \boldsymbol{\nu}\right)\right] \\
=& \mathbb{E}\left[\ln p(\mathbf{Y}, \mathbf{S}|\mathbf{X}, \boldsymbol{\Phi}, \boldsymbol{\Psi})\right] - \mathbb{E}\left[\ln q\left(\mathbf{S}\right)\right] \\
& + \mathbb{E}\left[\ln p(\boldsymbol{\Phi}|\boldsymbol{\lambda})\right] + \mathbb{E}\left[\ln p(\boldsymbol{\Psi}|\boldsymbol{\nu})\right] + \mathbb{E}\left[\ln p(\boldsymbol{\lambda})\right] + \mathbb{E}\left[\ln p(\boldsymbol{\nu})\right] \\
& - \mathbb{E}\left[\ln q\left(\boldsymbol{\Phi}\right)\right] - \mathbb{E}\left[\ln q\left(\boldsymbol{\Psi}\right)\right] - \mathbb{E}\left[\ln q\left(\boldsymbol{\lambda}\right)\right] - \mathbb{E}\left[\ln q\left(\boldsymbol{\nu}\right)\right]
\end{aligned}
\tag{35}
$$

where all expectations are taken with respect to the approximate posteriors of their arguments and the superscript $\star$ is omitted. It can be seen that the expectations of the distributions $q$ are expectations of their log with respect to themselves (negative entropies).

We will only show expectations which can be generally derived for any type of HMDT model. The expectations which are specific to the regression problem or the used prior will be derived in a later section.

$$
\begin{aligned}
\mathbb{E}\left[\ln q\left(\mathbf{S}\right)\right] =& \sum_{\mathbf{S}} q\left(\mathbf{S}\right) \ln q\left(\mathbf{S}\right) \\
=& \sum_{\mathbf{S}} q\left(\mathbf{S}\right) \left(\mathbb{E}_{q(\boldsymbol{\Phi})q(\boldsymbol{\Psi})}\left[\ln p(\mathbf{Y}, \mathbf{S}|\mathbf{X}, \boldsymbol{\Phi}, \boldsymbol{\Psi})\right] - \ln \tilde{\mathcal{Z}}\left(\mathbf{Y}|\mathbf{X}\right)\right) \\
=& \sum_{\mathbf{S}} q\left(\mathbf{S}\right) \left(\mathbb{E}_{q(\boldsymbol{\Phi})q(\boldsymbol{\Psi})}\left[\ln p(\mathbf{Y}, \mathbf{S}|\mathbf{X}, \boldsymbol{\Phi}, \boldsymbol{\Psi})\right]\right) - \ln \tilde{\mathcal{Z}}\left(\mathbf{Y}|\mathbf{X}\right) \\
=& \mathbb{E}\left[\ln p(\mathbf{Y}, \mathbf{S}|\mathbf{X}, \boldsymbol{\Phi}, \boldsymbol{\Psi})\right] - \ln \tilde{\mathcal{Z}}\left(\mathbf{Y}|\mathbf{X}\right)
\end{aligned}
\tag{36}
$$

where we see that

$$
\ln \tilde{\mathcal{Z}}\left(\mathbf{Y}|\mathbf{X}\right) = \mathbb{E}\left[\ln p(\mathbf{Y}, \mathbf{S}|\mathbf{X}, \boldsymbol{\Phi}, \boldsymbol{\Psi})\right] - \mathbb{E}\left[\ln q\left(\mathbf{S}\right)\right]
\tag{37}
$$

which is the normalisation constant we calculate during the forward-back step (see above). Therefore, the lower bound reduces to

$$
\begin{aligned}
\mathcal{L} =& \ln \tilde{\mathcal{Z}}\left(\mathbf{Y}|\mathbf{X}\right) \\
& + \mathbb{E}\left[\ln p(\boldsymbol{\Phi}|\boldsymbol{\lambda})\right] + \mathbb{E}\left[\ln p(\boldsymbol{\Psi}|\boldsymbol{\nu})\right] + \mathbb{E}\left[\ln p(\boldsymbol{\lambda})\right] + \mathbb{E}\left[\ln p(\boldsymbol{\nu})\right] \\
& - \mathbb{E}\left[\ln q\left(\boldsymbol{\Phi}\right)\right] - \mathbb{E}\left[\ln q\left(\boldsymbol{\Psi}\right)\right] - \mathbb{E}\left[\ln q\left(\boldsymbol{\lambda}\right)\right] - \mathbb{E}\left[\ln q\left(\boldsymbol{\nu}\right)\right].
\end{aligned}
\tag{38}
$$

## 2 Bayesian Logistic Regression

Logistic regression is a common choice for modelling conditional probabilities of a target variable which follows a Bernoulli distribution. It takes the form of

$$p\left(y=1|\mathbf{x},\mathbf{w}\right) = \sigma\left(\mathbf{w}^{\mathsf{T}}\mathbf{x}\right) = \frac{1}{1+\exp\left(-\mathbf{w}^{\mathsf{T}}\mathbf{x}\right)} \tag{39}$$

where $y$ is the target variable with $y \in \{0, 1\}$, $\mathbf{x}$ is a vector of covariates and $\mathbf{w}$ are the linear weights for each covariate. $\sigma\left(a\right) \in [0, 1]$ is called the logistic sigmoid which is symmetric around zero such that

$$\sigma\left(a\right) = 1 - \sigma\left(-a\right). \tag{40}$$

From this property, it can be shown that the conditional probability can be written as (see [3, Ch. 10.6.1])

$$
\begin{aligned}
p\left(y|\mathbf{w},\mathbf{x}\right) &= \sigma\left(\mathbf{w}^{\mathsf{T}}\mathbf{x}\right)^{y}\left(1-\sigma\left(\mathbf{w}^{\mathsf{T}}\mathbf{x}\right)\right)^{(1-y)} \\
&= \exp\left(\left(\mathbf{w}^{\mathsf{T}}\mathbf{x}\right)y\right)\sigma\left(-\mathbf{w}^{\mathsf{T}}\mathbf{x}\right)
\end{aligned} \tag{41}
$$

The logistic regression is a generalised linear model which makes model parameter inference simple if we want to perform maximum likelihood estimation. However, for a Bayesian treatment the posterior distribution over the parameters of a logistic regression is intractable.

To overcome this problem we will consider the approach by [9] for performing Bayesian logistic regression. The authors introduce a variational lower bound to the logistic sigmoid which is quadratic in its exponent. This enables us to estimate the posterior distribution in closed form for parameters with a Gaussian prior. The descriptions on this approach will be kept to a minimum in this section. The interested reader is referred to [9] and [3, Ch. 10.5 & 10.6].

The lower bound on the logistic sigmoid can be written as

$$\sigma\left(a\right) \geq \sigma\left(\xi\right)\exp\left(\left(a-\xi\right)/2 - \lambda\left(\xi\right)\left(a^{2}-\xi^{2}\right)\right) \tag{42}$$

where

$$\lambda\left(\xi\right) = \frac{1}{4\xi}\tanh\left(\frac{\xi}{2}\right) \tag{43}$$

and $\xi$ is a local variational parameter which is specific to any single observation. By substituting this equation into 41 the Bernoulli likelihood can be lower bounded by

$$p\left(t|\mathbf{x},\mathbf{w}\right) \geq \exp\left(\left(\mathbf{w}^{\mathsf{T}}\mathbf{x}\right)y\right)\sigma\left(\xi\right)\exp\left(-\left(\mathbf{w}^{\mathsf{T}}\mathbf{x}+\xi\right)/2 - \lambda\left(\xi\right)\left(\left(\mathbf{w}^{\mathsf{T}}\mathbf{x}\right)^{2}-\xi^{2}\right)\right) = p\left(y|\mathbf{x},\mathbf{w},\xi\right) \tag{44}$$

where we refer to the lower bound on the likelihood as $p\left(y|\mathbf{x},\mathbf{w},\xi\right)$ because we introduced the additional parameter $\xi$.

### 2.1 Parameter priors

In order to perform Bayesian inference on the logistic regression, we define Gaussian priors over the parameters of the logistic regression. For each parameter we assume independent univariate Gaussian priors with zero mean. Furthermore, we introduce hyperpriors over the precision of each Gaussian prior. This enables us to define an anisotropic prior over the model parameters allowing us to perform automatic relevance determination (ARD) [10]. In the following section we will focus on the ARD prior. For derivations using an isotropic prior the reader is referred to [3, Ch. 10.6].

The anisotropic Gaussian prior takes the form

$$p(\mathbf{w}|\boldsymbol{\alpha}) = \prod_{d=1}^{D}\mathcal{N}(w_d|0,\alpha_d^{-1}) = \frac{|\mathbf{A}|^{-1/2}}{\sqrt{2\pi}^D}\exp\left(-\frac{1}{2}\mathbf{w}^{\mathsf{T}}\mathbf{A}\mathbf{w}\right) \tag{45}$$

where $D$ is the number of parameters and $\mathbf{A}$ is a diagonal matrix with $\mathbf{A}_{dd} = \alpha_d^{-1}$.

We define the hyperpriors over the precision of the Gaussian prior as

$$p(\boldsymbol{\alpha}) = \prod_{d=1}^{D} \mathsf{Gam}(\alpha_d | \mathrm{a}_d^{(0)}, \mathrm{b}_d^{(0)}) \quad = \prod_{d=1}^{D} \frac{\left(\mathrm{b}_d^{(0)}\right)^{\mathrm{a}_d^{(0)}}}{\Gamma\left(\mathrm{a}_d^{(0)}\right)} \alpha_d^{\mathrm{a}_d^{(0)}-1} \exp\left(-\mathrm{b}_d^{(0)} \alpha_d\right), \qquad (46)$$

where $\mathrm{a}_d^{(0)} > 0$ and $\mathrm{b}_d^{(0)} > 0$ are shape and rate parameters which are initially set. Although it will not be described in this section it is possible to also perform inference over those parameters via gradient or sampling methods.

## 2.2 Update equations

To derive the variational posterior we can use the result from 34 such that the log-factor $q^\star(\mathbf{w})$ has the form

$$
\begin{aligned}
\ln q^\star(\mathbf{w}) &= \sum_{t=0}^{T-1} \eta_t \ln p\left(y_t | \mathbf{x}_t, \mathbf{w}\right) + \mathbb{E}_{\boldsymbol{\alpha}}\left[\ln p(\mathbf{w}|\boldsymbol{\alpha})\right] + \text{const.} \\
&\geq \sum_{t=0}^{T-1} \eta_t \ln p\left(y_t | \mathbf{x}_t, \mathbf{w}, \xi_t\right) + \mathbb{E}_{\boldsymbol{\alpha}}\left[\ln p(\mathbf{w}|\boldsymbol{\alpha})\right] + \text{const.} \\
&= \sum_{t=0}^{T-1} \eta_t \ln \ln p\left(y_t | \mathbf{x}_t, \mathbf{w}, \xi_t\right) - \frac{1}{2} \sum_{d=1}^{D} \mathbb{E}_{\boldsymbol{\alpha}}\left[\alpha_d\right] w_d^2 + \text{const.} \\
&= \sum_{t=0}^{T-1} \eta_t \left(\ln \sigma\left(\xi_t\right) - \frac{\xi_t}{2} + \lambda(\xi_t)\xi_t^2 + \mathbf{w}^\intercal \mathbf{x}_t (y_t - 0.5) - \lambda\left(\xi_t\right) \mathbf{w}^\intercal \mathbf{x}_t \mathbf{x}_t^\intercal \mathbf{w}\right) \\
&\quad - \frac{1}{2} \mathbf{w}^\intercal \mathbb{E}_{\boldsymbol{\alpha}}\left[\mathbf{A}\right] \mathbf{w} + \text{const.} \\
&= \mathbf{w}^\intercal \sum_{t=0}^{T-1} \eta_t \mathbf{x}_t \left(y_t - 0.5\right) - \frac{1}{2} \mathbf{w}^\intercal \left(\mathbb{E}_{\boldsymbol{\alpha}}\left[\mathbf{A}\right] + 2 \sum_{t=0}^{T-1} \eta_t \lambda\left(\xi_t\right) \mathbf{x}_n \mathbf{x}_n^\intercal\right) \mathbf{w} + \text{const.}
\end{aligned}
\tag{47}
$$

where the first line of this equation is as 34. We have brought $\ln q^\star(\mathbf{w})$ into a quadratic function of $\mathbf{w}$. By reading out the linear and quadratic terms in $\mathbf{w}$ we find the mean and and covariance of the Gaussian posterior

$$q^\star(\mathbf{w}) = \mathcal{N}\left(\boldsymbol{\mu}, \boldsymbol{\Sigma}\right) \tag{48}$$

with

$$\boldsymbol{\Sigma}^{-1} = \mathbb{E}_{\boldsymbol{\alpha}}\left[\mathbf{A}\right] + 2 \sum_{n=1}^{N} \eta_n \lambda\left(\xi_n\right) \mathbf{x}_n \mathbf{x}_n^\intercal \tag{49}$$

$$\boldsymbol{\mu} = \boldsymbol{\Sigma} \sum_{n=1}^{N} \eta_n \mathbf{x}_n \left(y_n - 0.5\right) \tag{50}$$

and $\mathbb{E}_{\boldsymbol{\alpha}}\left[\mathbf{A}\right]$ is a diagonal matrix with $\mathbb{E}\left[\alpha_d\right]$ as elements.

We now derive the approximate posterior over the hyperparameters where we again start by taking the the expectation of the logarithm of the joint distribution. By moving terms independent of $\boldsymbol{\alpha}$ into the constant term we find

$$
\begin{aligned}
\ln q^\star(\boldsymbol{\alpha}) &= \mathbb{E}_{\mathbf{w}}\left[\ln p(\mathbf{w}|\boldsymbol{\alpha}^{-1})\right] + \ln p\left(\boldsymbol{\alpha}\right) + \text{const.} \\
&= \sum_{d=1}^{D} \left(\mathbb{E}_{w_d}\left[\ln p(w_d|\alpha_d^{-1})\right] + \ln p\left(\alpha_d\right)\right) + \text{const.}
\end{aligned}
\tag{51}
$$

where we made use of 46. It can be seen that $\ln q^\star(\boldsymbol{\alpha})$ factorises into $D$ independent components. We can therefore derive the posterior for each $\alpha_d$ by

$$
\begin{aligned}
\ln q^\star(\alpha_d) &= \mathbb{E}_{w_d}\left[\ln p(w_d|\alpha_d^{-1})\right] + \ln p\left(\alpha_d\right) + \text{const.} \\
&= \int \left(\frac{1}{2}\ln \alpha_d - \frac{1}{2}\ln 2\pi - \frac{1}{2}\alpha_d w_d^2\right) q\left(w_d\right) \mathrm{d}w_d + (\mathrm{a}_d^{(0)} - 1)\ln \alpha_d - \mathrm{b}_d^{(0)}\alpha_d + \text{const.} \\
&= \frac{1}{2}\ln \alpha_d - \frac{1}{2}\alpha_d\left[\sigma_d^2 + \mu_d^2\right] + (\mathrm{a}_d^{(0)} - 1)\ln \alpha_d - \mathrm{b}_d^{(0)}\alpha_d + \text{const.} \\
&= \underbrace{(\mathrm{a}_d^{(0)} - \frac{1}{2})}_{(\mathrm{a}_d - 1)}\ln \alpha_d - \underbrace{\left[\mathrm{b}_d^{(0)} + \frac{1}{2}\left[\sigma_d^2 + \mu_d^2\right]\right]}_{\mathrm{b}_d}\alpha_d + \text{const.} \\
&= \ln \mathsf{Gam}\left(\alpha_d | \mathrm{a}_d, \mathrm{b}_d\right).
\end{aligned}
\tag{52}
$$

which takes the form of the logarithm of a Gamma distribution. We come to these results via the conjugate-exponential relationship of the Gaussian distribution and the Gamma distribution. The corresponding shape and rate parameters can be read out with

$$
\mathrm{a}_d = \mathrm{a}_d^{(0)} + \frac{1}{2}
\tag{53}
$$

and

$$
\mathrm{b}_d = \mathrm{b}_d^{(0)} + \frac{1}{2}\left[\sigma_d^2 + \mu_d^2\right]
\tag{54}
$$

The mean of the gamma distribution is given by

$$
\mathbb{E}\left[\alpha_d\right] = \frac{\mathrm{a}_d}{\mathrm{b}_d}
\tag{55}
$$

The expectation of the log-probability with respect to $q(\mathbf{w})$ is evaluated by

$$
\begin{aligned}
&\mathbb{E}_{\mathbf{w}}\left[\ln p\left(y_n|\mathbf{x}_n, \mathbf{w}, \xi_n\right)\right] \\
&= \mathbb{E}_{\mathbf{w}}\left[\mathbf{w}^{\mathsf{T}}\mathbf{x}_n(y_n - 0.5) - \lambda(\xi_n)\mathbf{w}^{\mathsf{T}}\mathbf{x}_n\mathbf{x}_n^{\mathsf{T}}\mathbf{w}\right] + \ln \sigma(\xi_n) - \frac{\xi_n}{2} + \lambda(\xi_n)\xi_n^2 \\
&= \boldsymbol{\mu}^{\mathsf{T}}\mathbf{x}_n(y_n - 0.5) - \lambda(\xi_n)\left(\mathbf{x}_n^{\mathsf{T}}E_{\mathbf{w}}[\mathbf{w}\mathbf{w}^{\mathsf{T}}]\mathbf{x}_n\right) + \ln \sigma(\xi_n) - \frac{\xi_n}{2} + \lambda(\xi_n)\xi_n^2 \\
&= \boldsymbol{\mu}^{\mathsf{T}}\mathbf{x}_n\left(y_n - 0.5\right) - \lambda\left(\xi_n\right)\left(\mathbf{x}_n^{\mathsf{T}}\left(\Sigma + \boldsymbol{\mu}\boldsymbol{\mu}^{\mathsf{T}}\right)\mathbf{x}_n\right) + \ln \sigma(\xi_n) - \frac{\xi_n}{2} + \lambda(\xi_n)\xi_n^2
\end{aligned}
\tag{56}
$$

This function is used for calculating the expectations necessary to perform the forward-backward algorithm.

### 2.2.1 Optimising the variational parameters

The variational parameters are updated by maximising the variational lower bound with respect to the parameters. The lower bound is only dependent on

$$
\begin{aligned}
\tilde{\mathcal{L}}\left(q, \boldsymbol{\xi}\right) &= \mathbb{E}_{\mathbf{Z}, \mathbf{w}}\left[\ln p(\mathbf{t}|\boldsymbol{\eta}, \mathbf{X}, \mathbf{w})\right] + \text{const.} \\
&= \iint \sum_{t=0}^{T-1} \eta_t \ln p\left(y_t|\mathbf{x}_t, \mathbf{w}\right) \mathrm{d}\boldsymbol{\eta}\mathrm{d}\mathbf{w} + \text{const.} \\
&= \sum_{t=0}^{T-1} \eta_t\mathbb{E}_{\mathbf{w}}\left[\ln p\left(y_t|\mathbf{x}_t, \mathbf{w}\right)\right] + \text{const.} \\
&= \sum_{t=0}^{T-1} \eta_t\left(-\lambda\left(\xi_t\right)\left(\mathbf{x}_t^{\mathsf{T}}\left(\Sigma + \boldsymbol{\mu}\boldsymbol{\mu}^{\mathsf{T}}\right)\mathbf{x}_t\right) + \ln \sigma(\xi_t) - \frac{\xi_t}{2} + \lambda(\xi_t)\xi_t^2\right) + \text{const.}
\end{aligned}
\tag{57}
$$

This factorises into $t$ parts for each variational parameter. Taking the partial derivative with respect to $\xi_t$ we get

$$\frac{\partial \tilde{\mathcal{L}}(q,\boldsymbol{\xi})}{\partial \xi_t} = \eta_t \frac{\partial}{\partial \xi_t}\left(-\lambda(\xi_t)(\mathbf{x}_t^\mathsf{T}(\Sigma + \boldsymbol{\mu}\boldsymbol{\mu}^\mathsf{T})\mathbf{x}_t) + \ln\sigma(\xi_t) - \frac{\xi_t}{2} + \lambda(\xi_n)\xi_t^2\right). \tag{58}$$

By setting this function to zero, and by solving for $\xi_t$ we get (see [3, Ch. 10.6.2]):

$$(\xi_t^{new})^2 = \mathbf{x}_t^\mathsf{T}(\Sigma + \boldsymbol{\mu}\boldsymbol{\mu}^\mathsf{T})\mathbf{x}_t \tag{59}$$

for the re-estimation of $\xi_t$.

## 2.3  Variational Lower Bound

As described earlier, the variational lower bound can be used for model comparison, to test convergence and as a sanity check. Below we calculate the expectations need for calculating the lower bound in equation (38):

$$
\begin{aligned}
\mathbb{E}_{\mathbf{w},\boldsymbol{\alpha}}[\ln p(\mathbf{w}|\boldsymbol{\alpha})] &= \mathbb{E}_{\mathbf{w},\boldsymbol{\alpha}}\left[\sum_{d=1}^{D} -\frac{1}{2}\ln\alpha_d - \frac{1}{2}\ln(2\pi) - \frac{1}{2}\alpha_d w_d^2\right]\\
&= -\frac{D_k}{2}\ln(2\pi) + \sum_{d=1}^{D}\left(-\frac{1}{2}\mathbb{E}[\ln\alpha_d] - \frac{1}{2}\mathbb{E}[\alpha_d]\,\mathbb{E}[w_d^2]\right)\\
&= -\frac{D}{2}\ln(2\pi) + \sum_{d=1}^{D}\left(-\frac{1}{2}(\psi(\mathrm{a}_d) - \ln \mathrm{b}_d) - \frac{1}{2}\frac{\mathrm{a}_d}{\mathrm{b}_d}(\sigma_d^2 + \mu_d^2)\right)
\end{aligned}
\tag{60}
$$

$$
\begin{aligned}
\mathbb{E}_{\boldsymbol{\alpha}}[\ln p(\boldsymbol{\alpha})] &= \mathbb{E}_{\boldsymbol{\alpha}}\left[\sum_{d=1}^{D} -\ln\Gamma\left(\mathrm{a}_d^{(0)}\right) + \mathrm{a}_d^{(0)}\ln \mathrm{b}_d^{(0)} + \left(\mathrm{a}_d^{(0)} - 1\right)\ln\alpha_d - \mathrm{b}_d^{(0)}\alpha_d\right]\\
&= \sum_{d=1}^{D} -\ln\Gamma\left(\mathrm{a}_d^{(0)}\right) + \mathrm{a}_d^{(0)}\ln \mathrm{b}_d^{(0)} + \left(\mathrm{a}_d^{(0)} - 1\right)\mathbb{E}[\ln\alpha_d] - \mathrm{b}_d^{(0)}\mathbb{E}[\alpha_d]\\
&= \sum_{d=1}^{D} -\ln\Gamma\left(\mathrm{a}_d^{(0)}\right) + \mathrm{a}_d^{(0)}\ln \mathrm{b}_d^{(0)} + \left(\mathrm{a}_d^{(0)} - 1\right)(\psi(\mathrm{a}_d) - \ln \mathrm{b}_d) - \mathrm{b}_d^{(0)}\frac{\mathrm{a}_d}{\mathrm{b}_d}
\end{aligned}
\tag{61}
$$

$$
\begin{aligned}
\mathbb{E}_{\mathbf{w}}[\ln q(\mathbf{w})] &= \mathbb{E}_{\mathbf{w}}\left[-\frac{1}{2}\ln|\Sigma| - \frac{D}{2}\ln(2\pi) - \frac{1}{2}(\mathbf{w} - \boldsymbol{\mu})^\mathsf{T}\Sigma^{-1}(\mathbf{w} - \boldsymbol{\mu})\right]\\
&= -\frac{1}{2}\ln|\Sigma| - \frac{D}{2}\ln(2\pi) - \frac{1}{2}\underbrace{\mathbb{E}_{\mathbf{w}}\left[(\mathbf{w} - \boldsymbol{\mu})^\mathsf{T}\Sigma^{-1}(\mathbf{w} - \boldsymbol{\mu})\right]}_{D}\\
&= -\frac{1}{2}\ln|\Sigma| - \frac{D}{2}\ln(2\pi) - \frac{1}{2}D\\
&= -\frac{1}{2}\ln|\Sigma| - \frac{D}{2}(1 + \ln(2\pi))
\end{aligned}
\tag{62}
$$

Figure 2: **Graphical model for hierarchical Bayesian inference with Gaussian priors on the parameter means as described in the text.** Here $\boldsymbol{m}$ and $\boldsymbol{w}_i$ are prior distributions for the mean of their children, $\boldsymbol{\alpha}_i$ and $\boldsymbol{\beta}$ are independent Gamma priors on the precisions of their respective children (Gaussians) and $\mu$ is the prior mean for $\boldsymbol{m}$ which can be either a Gaussian again or a constant. This hierarchical structure can be extended further upwards dependent on the depth of the HMDT.

$$
\begin{aligned}
\mathbb{E}_{\boldsymbol{\alpha}}\left[\ln q(\boldsymbol{\alpha})\right] &= \mathbb{E}_{\boldsymbol{\alpha}}\left[\sum_{d=1}^{D} -\ln \Gamma\left(\mathrm{a}_d\right) + \mathrm{a}_d \ln \mathrm{b}_d + \left(\mathrm{a}_d - 1\right)\ln \alpha_d - \mathrm{b}_d \alpha_d\right] \\
&= \sum_{d=1}^{D} -\ln \Gamma\left(\mathrm{a}_d\right) + \mathrm{a}_d \ln \mathrm{b}_d + \left(\mathrm{a}_d - 1\right)\mathbb{E}\left[\ln \alpha_d\right] - \mathrm{b}_d \mathbb{E}\left[\alpha_d\right] \\
&= \sum_{d=1}^{D} -\ln \Gamma\left(\mathrm{a}_d\right) + \mathrm{a}_d \ln \mathrm{b}_d + \left(\mathrm{a}_d - 1\right)\left(\psi\left(\mathrm{a}_d\right) - \ln \mathrm{b}_d\right) - \mathrm{b}_d \frac{\mathrm{a}_d}{\mathrm{b}_d} \\
&= \sum_{d=1}^{D} -\ln \Gamma\left(\mathrm{a}_d\right) + \left(\mathrm{a}_d - 1\right)\psi\left(\mathrm{a}_d\right) - \ln \mathrm{b}_d - \mathrm{a}_d
\end{aligned}
\tag{63}
$$

where $\psi(...)$ is the digamma function.

## 3 Hierarchical Bayesian inference

We start from a situation with I independent logistic regressions $p_i\left(y|\mathbf{w}_i, \mathbf{x}\right)$. Say those regressions model the spiking probability for a neuron in I states. In the following description we will assume that the regression parameters for each state are similar. This way it is possible to model prior assumptions about similar receptive field for state close-by in the HMDT. To implement such an assumption we chose a common mean (instead of zero mean) for the parameter priors such that the anisotropic Gaussian from earlier takes the form

$$
p(\mathbf{w}_i|\mathbf{m}, \boldsymbol{\alpha}_i) = \prod_{d=1}^{D} \mathcal{N}(w_{i,d}|m_d, \alpha_{i,d}^{-1}) = \frac{|\mathbf{A}_i|^{-1/2}}{\sqrt{2\pi}^D} \exp\left(-\frac{1}{2}(\mathbf{w}_i - \mathbf{m})^{\mathsf{T}}\mathbf{A}_i(\mathbf{w}_i - \mathbf{m})\right),
\tag{64}
$$

where the Gamma prior on the precision takes the same form as before:

$$
p(\boldsymbol{\alpha}_i) = \prod_{d=1}^{D} \mathsf{Gam}(\alpha_{i,d}|\mathrm{a}_{i,d}^{(0)}, \mathrm{b}_{i,d}^{(0)}) = \prod_{d=1}^{D} \frac{\left(\mathrm{b}_{i,d}^{(0)}\right)^{\mathrm{a}_{i,d}^{(0)}}}{\Gamma\left(\mathrm{a}_{i,d}^{(0)}\right)} \alpha_{i,d}^{\mathrm{a}_{i,d}^{(0)}-1} \exp\left(-\mathrm{b}_{i,d}^{(0)}\alpha_{i,d}\right)
\tag{65}
$$

We will further assume that the common prior mean $\mathbf{m}$ from above follows a Gaussian distribution. Bayesian inference for this structure is straightforward because the Gaussian is

the conjugate prior for the mean of a Gaussian [3]. Similarly to before the prior distribution over $\mathbf{m}$ takes the form of an anisotropic Gaussian of the form

$$p(\mathbf{m}|\boldsymbol{\mu},\boldsymbol{\beta}) = \prod_{d=1}^{D} \mathcal{N}(m_d^{(i)}|\mu_d,\beta_d^{-1}) = \frac{|\mathbf{B}|^{-1/2}}{\sqrt{2\pi}^D} \exp\left(-\frac{1}{2}\left(\mathbf{m}-\boldsymbol{\mu}\right)^{\mathsf{T}}\mathbf{B}\left(\mathbf{m}-\boldsymbol{\mu}\right)\right) \tag{66}$$

with independent Gamma priors on it's precision matrix:

$$p(\boldsymbol{\beta}) = \prod_{d=1}^{D} \mathsf{Gam}(\beta_d|c_d^{(0)},d_d^{(0)}) \quad = \prod_{d=1}^{D} \frac{\left(d_d^{(0)}\right)^{c_d^{(0)}}}{\Gamma\left(c_d^{(0)}\right)} \beta_d^{c_d^{(0)}-1} \exp\left(-d_d^{(0)}\beta_d\right) \tag{67}$$

This hierarchical structure can be further extended by defining a prior over $\boldsymbol{\mu}$. A graphical model to show the hierarchical Bayesian network is depicted in figure 2. From this representation it should become clear how the HMDT can give rise to a hierarchical prior structure. States in the same sub-tree of the HMDT share the same prior mean, this prior mean follows a distribution which has a shared mean with states further away in the tree, and so on.

The resulting update equation for the posterior distributions over parameters are given by

$$\boldsymbol{\Sigma}_{ji}^{-1} = \mathbb{E}_{\boldsymbol{\alpha}_i}\left[\mathbf{A}_i\right] + 2\sum_{n=1}^{N} \eta_n \lambda\left(\xi_n\right)\mathbf{x}_n\mathbf{x}_n^{\mathsf{T}} \tag{68}$$

$$\boldsymbol{\mu}_{\mathbf{i}} = \boldsymbol{\Sigma}_{\mathbf{i}}\left[\mathbb{E}_{\boldsymbol{\alpha}_i}\left[\mathbf{A}_i\right]\boldsymbol{\mu}_{\mathbf{m}} + \sum_{n=1}^{N} \eta_n\mathbf{x}_n\left(y_n - 0.5\right)\right] \tag{69}$$

for the regression parameters and by

$$\boldsymbol{\Sigma}_{\mathbf{m}}^{-1} = \mathbb{E}_{\boldsymbol{\beta}}\left[\mathbf{B}\right] + \sum_{i=1}^{I} \mathbb{E}_{\boldsymbol{\alpha}_i}\left[\mathbf{A}_i\right] \tag{70}$$

$$\boldsymbol{\mu}_{\mathbf{m}} = \boldsymbol{\Sigma}_{\mathbf{m}}\left[\mathbb{E}_{\boldsymbol{\beta}}\left[\mathbf{B}\right]\mathbb{E}_{\boldsymbol{\mu}}\left[\boldsymbol{\mu}\right] + \sum_{i=1}^{I} \mathbb{E}_{\boldsymbol{\alpha}_i}\left[\mathbf{A}_i\right]\boldsymbol{\mu}_i\right] \tag{71}$$

for the higher level prior parameters. Posterior inference over the precision parameters $\boldsymbol{\alpha}_o$ and $\boldsymbol{\beta}$ works as above. Those posterior distributions can be alternatively updated as part of the VBEM algorithm.