[Reviews · NeurIPS 2014]

Submitted by Assigned_Reviewer_26

This manuscript presents a flexible discrete latent-state model for population neural data. Approximations (variational and eq 10,11) are necessary to do inference in powerful flexible model. This is a tool for confirmatory analysis; one major weakness as a explorative tool is the necessity to set up the state hierarchy in advance.

Originality: It is a novel approach.

Quality & Clarity:
Some critical details such as the number of neurons of the V1 data are missing.
There are lots of details in the supplement, but I have not checked those.

Significance:
It is a powerful discrete state latent variable model. Results look interesting.

Other comments:
The hierarchical organization of states is presumably to make the parameterization simpler as well, but this is not addressed. Is the hierarchy interpretable otherwise? The motivation and consequence of using this particular form needs more explanation.

What's the ML fit in Fig 2D? Was it EM?

When comparing to coupled GLM (I assumed that's the 2-state GLM in the figure? it's not clear from the text. why is that 2-state anyways?), how were the temporal filters parameterized (and the bin size for Fig 3)? This could be critical factor for achieving a good fit.

Author feedback:
Thank you for explaining the details. Please make sure to update the final paper.
Summary: Proposed model has potential to significantly improve neural population analysis. Some details were missing, but rebuttal is satisfactory.

Submitted by Assigned_Reviewer_27

The firing properties of neurons in the mammalian brain are not constant over time. One kind of non-stationarity that has been observed is that under certain conditions, populations of neurons can switch between discrete states, across which their inputs and responsiveness can vary dramatically. In turn, there may be structure to this network state-space. Despite all this, much neurophysiological data has been analysed whilst ignoring these discrete spaces, or with hacky/ad-hoc approaches to separate out the states. This paper attempts to tackle such population datasets by integrating GLM-style modelling of stimulus-response relationships into a hierarchical discrete state-space framework. The authors construct a statistical model for neural responses under this framework, based on a hierarchical HMM (i.e., an HME), and approximate inference under it using VBEM. The authors then demonstrate its recovery of ground truth under a simulated dataset, and its application to real neural data.

Quality: The statistical model and its inference appear clean and sound, and for the examples presented, it performs marginally (but significantly) better on both simulated and real data compared with less complex models (e.g. when priors are removed; or using a two-state rather than a hierarchical three state model). Having said that, while it is certainly an improvement over previously presented models within the same class (e.g. Escola et al (2011), Chen et al (2009)), the improvement in performance is pretty small. Moreover, there does not appear to be a strong argument that the *hierarchical* nature of this model actually adds anything. There is no explicit comparison with an equally powerful non-hierarchical model (e.g. 3-state HMMs with VB for the state model, to compare with the examples of 3-state HMEs). This leaves me scratching my head, since, according to the title, the hierarchical aspect of the state space is the novel component of this model. Without seeing the data, I worry a bit about whether this promise can be achieved on the real datasets in question, since the improvement of 3-state HMEs over 2-state HMMs is typically small (e.g. Fig 3C, Fig 4 caption). Of course, the authors can convince me otherwise by presenting the results, in which case I would give the paper score a bump. Otherwise, while I can appreciate the promise of this approach, the real data may not justify such a model. For instance, the "hierarchy of discrete states" may be better described by a continuous latent factor. I'm nevertheless in favour of acceptance since this work is sound and should be published, but its scope of utility may not yet have been realised.

Clarity: The paper is clear and well-written. Some minor points:

- line 51: "simple a global" -> "a simple global"
- line 99: "as well as hierarchical relationships between trials". Not sure what this is meant to mean. The architecture of the HMDT is fixed (though its parameters are learned). What does "between trials" mean here?
- line 106: abnormal sentence termination "non-geometrically".
- Fig 1B: some captioning/labelling of the transition matrices is necessary -- I couldn't parse these
- line 112: "ypically"
- line 267: "where theta is the phase" -> should be phi
- line 267: pairwise product -> outer product?
- line 266-268: a little motivation for the chosen parameterisation of the stimulus would be useful (yes, it's tangential to the core of the paper, but it's a bit bewildering. These look like a filtering of the stimulus by a set of 15 direction-selective/non-selective and orientation-selective/non-selective simple cells, yes?)
- line 321: demote Macaques to plain old macaques
- Fig 4: not referenced in main paper
- line 386: "for the characterized..."?
- line 406: "both, better" omit comma
- the British author and American author should wrestle it out to determine spelling conventions

Originality: This is an improved statistical model above and beyond other recently published work. However, as mentioned above, I think a full comparison is necessary to fully evaluate this work.

Significance: See comments above.
Summary: The paper presents an improved model for analysing data from particular non-stationarity neural datasets. The demonstrated performance boost is only small though, and I'd like to see a fair comparison to justify the use of the word "hierarchically".

Submitted by Assigned_Reviewer_32

The authors used hierarchical hidden markov model to describe how the populational state of a neural system affect the fluctuations of activities in local cortical circuits. They used markov decision tree structure to model the latent "gate" variables, which in turn statistically determine the population states. For efficient estimation of the parameters and hyper-parameters of this hierarchical model, the authors applied Variational Bayesian approximation to the posterior distribution given observed neural activities and external variables. The resultant algorithm is a variation of EM algorithm. The significant merits of this framework were demonstrated by simulated as well as real experimental data.
Summary: This work has provided a novel, high-quality way to understand the population states of neural system. The clarity of this work is also superior, despite some minor typos.
Author Feedback
Author rebuttal: We thank the reviewers for their constructive feedback which prompted us to perform several new analyses, and their positive appraisal of our method. Both Rev 26 and 27 raised questions regarding the motivation and consequences of our ‘hierarchical organization’, which was insufficiently explained in the draft.

a) From a neuroscience perspective, our motivation comes from the fact that cortical states have been described to be hierarchically organized-- cortical activity can switch between desynchronized and synchronized states, synchronized states exhibit up and down phases, and the up-phases contain transient population bursts [8]. This structure naturally yields a 3-level binary tree: States can be divided into sub-classes, with states further down in the tree operating at faster time-scales determined by their parent node. We hypothesize that other cortical states also exhibit similar structure.

b) A hierarchical structure can yield more interpretable models. For example, it is reasonable to assume that neurons have similar firing properties in similar states. This assumption can be enforced using a prior derived from the tree structure, in which states close to each other in the tree are enforced to have similar parameters.

c) The hierarchical structure has algorithmic advantages over to a flat one (as hypothesized by rev 26). Our model is based on a product of binary logistic regressions, each of which can be approximated using the bound from [21]. In a flat model we would need to use of a multinomial logistic regression, for which a bound was proposed by Bouchard (2007 NIPS workshop). It requires a second variational parameter, and it is not clear how to update those (interdependent) parameters. To the best of our knowledge--and certainly within the context of neuroscience---this approach has not been used for VB for HMMs with GLM observations. Therefore, the suggestion by reviewer 27 would be a comparison between two novel methods. Finally, our model-setup is valid also for multinomial gates.

d) In general, the hierarchical approach can model richer dependence of states on external covariates (analogous to the difference between multi-class logistic regression and multi-class binary decision trees). For example, a two-level binary model (but not 4-class logistic regression) can separate four point clouds situated at the corners of a square. However, due to our use of simple features for the gates (binary) and the simple structure (3 states), we would not expect an advantage of the hierarchical over the flat model for the examples used in the paper. For the data in the paper, the hierarchy aids interpretability and algorithmic tractability, but would not imply a superior likelihood compared to a flat model (and indeed we do not claim this in the paper).

Based on the reviewer’s suggestion, we implemented ML-inference for a flat 3-state model (note that VB for the flat model is not straightforward), and indeed it performs on par with the ML hierarchical 3-state model, but worse than our approach. We will include this comparison and clarify that it is due to the advantage brought about by VB (and not by hierarchical vs flat).

Reviewer 26

i) “not useful for exploration as hierarchy needs to be set in advance”:
The method can be used for exploratory analyses: If the model is run with a large tree, our VB-method aggressively only uses those states that are necessary to explain the data. Based on the reviewers suggestion, we now ran a model with a 3 level tree, i.e. 8 potential states on our simulated data, and found that the best model fit only used 3 of those 8 states (the other 5 states occupied less than 0.5%). An alternative approach would be to iteratively ‘grow’ the tree until the variational bound does not increase any more. Thus, our model can also be useful if the hierarchy is not known a priori.

ii) “comparing to coupled GLM”
Our one-state GLM is not coupled. Based on this suggestion, we fit a fully coupled GLM (with cross-history terms as in Pillow 2006), as well as one in which the total population count was used as a history feature, to our data. We found these models to be intermediate between the 1-state and 2-state GLM, i.e. both worse than the 3-state.

iii) “How were filters parameterized ... the number of neurons of V1 data are missing”
We apologize for an error in the labeling of figure 3D(v). The correct labeling should go from -50ms to -500ms. For the temporal filter we used 5 cubic B-splines with a logarithmic knot vector (10 bins length). We used a binning of 50ms. In the simulation we used 20 units, the real data 32 single cell units.

Reviewer 27

i) “since improvement of HME over 2-state HMMs is typically small (e.g. Fig 3C, Fig4 caption)”
- Figure 3C shows decoding performance-- while there is only a small gain in decoding, we note that the model is optimized for encoding, and it might just be the case that the additional state does not contain additional stimulus information. In addition, we now also plot the population spike histograms for the 2-state and 1-state model in Fig. 3G, and we find that both of them deviate substantially from the real data.

ii) Fig. 4: We agree, in this data set our model only finds evidence for 2 states, and we do not claim that in any population we find a particular number of states (but note that the VB-method for this model is still novel!). Our proposed model makes it possible to test hypotheses regarding the number and organisation of states.

iii) “the ‘discrete hierarchy’ may be better described by a continuous latent factor”
This is an intriguing question- whether the best descriptions of cortical states are multi-dimensional, discrete or continuous is an open question [8], and we hope that our method can help shed some light on these questions.
iv) Minor comments lines 51-406: Many thanks for these useful comments. We note that the British-writing author won.